# Structural basis of substrate recognition and translocation by human very long-chain fatty acid transporter ABCD1

Zhi-Peng Chen[1,2], Da Xu[1,2], Liang Wang[1,2], Yao-Xu Mao[1,2], Yang Li[1,2], Meng-Ting Cheng[1,2], Cong-Zhao Zhou [1,2✉], Wen-Tao Hou[1,2✉] & Yuxing Chen [1,2✉]

Human ABC transporter ABCD1 transports very long-chain fatty acids from cytosol to peroxisome for β-oxidation, dysfunction of which usually causes the X-linked adrenoleukodystrophy (X-ALD). Here, we report three cryogenic electron microscopy structures of ABCD1: the apo-form, substrate- and ATP-bound forms. Distinct from what was seen in the previously reported ABC transporters, the two symmetric molecules of behenoyl coenzyme A (C22:0-CoA) cooperatively bind to the transmembrane domains (TMDs). For each C22:0-CoA, the hydrophilic 3′-phospho-ADP moiety of CoA portion inserts into one TMD, with the succeeding pantothenate and cysteamine moiety crossing the inter-domain cavity, whereas the hydrophobic fatty acyl chain extends to the opposite TMD. Structural analysis combined with biochemical assays illustrates snapshots of ABCD1-mediated substrate transport cycle. It advances our understanding on the selective oxidation of fatty acids and molecular pathology of X-ALD.

[1] School of Life Sciences, University of Science and Technology of China, Hefei 230027, China. [2] The First Affiliated Hospital of USTC, Division of Life Sciences and Medicine, University of Science and Technology of China, Hefei 230027, China. ✉email: zcz@ustc.edu.cn; todvince@mail.ustc.edu.cn; cyxing@ustc.edu.cn

Fatty acids are critically important in cellular homeostasis for their functions in a variety of biological processes including energy storage and supply, phospholipid synthesis, protein post-translational modifications, cell signaling, membrane permeability and fluidity, and transcription control[1,2]. Human fatty acids are obtained through endogenous biosynthesis and dietary intake. To provide energy for the cell, the fatty acids subjected to oxidation are first activated by the acyl-CoA synthetase in the cytosol, forming fatty acyl-coenzyme A (fatty acyl-CoA). Afterwards, the short-chain (<C8) and medium-chain (C8–C14) fatty acyl-CoAs can directly diffuse into mitochondria, whereas long-chain (C16–C20) fatty acyl-CoAs (LCFA-CoAs) are transported to mitochondria via the carnitine shuttle[1,3]. In contrast, very long-chain (>C20) fatty acyl-CoAs (VLCFA-CoAs) and branched-chain fatty acyl-CoA are respectively transported into peroxisomes by three ATP-binding cassette (ABC) transporters, ABCD1/2/3, utilizing the energy of ATP hydrolysis[4–7]. Finally, various fatty acyl-CoAs are catabolized by a set of enzymes in either mitochondria or peroxisomes usually via β-oxidation[3,8].

The disruption of peroxisomal fatty acid catabolism is usually involved in the X-linked adrenoleukodystrophy (X-ALD), a progressive genetic disorder affecting the adrenal glands, spinal cord, and central nervous system. X-ALD is the most common peroxisomal disease with a prevalence of up to 1/20,000 in the adult male population[9,10]. In X-ALD patients, VLCFA-CoAs are unable to enter the peroxisome for β-oxidation, thus accumulated in the cytosol[5,11,12]. A series of reports indicated that X-ALD is caused by pathogenic variants in the *ABCD1* gene, which encodes the ABC transporter ABCD1 localized in the peroxisomal membrane, thus originally termed adrenoleukodystrophy protein[5,13,14].

Human ABCD1 belongs to the ABCD subfamily with four members ABCD1-4[13,15–17], which were classified into the type-IV ABC transporters[18]. As a half transporter, the ABCD transporter is expressed as a single polypeptide that contains a transmembrane domain (TMD) and a nucleotide-binding domain (NBD). ABCD1-4 usually function as homodimers[4,19,20], although heterodimers of ABCD1-3 have been also observed in some cases[21,22]. Distinct from the putative cobalamin transporter ABCD4 localized in the lysosomal membrane[23,24], ABCD1-3 are localized in the peroxisomal membrane[17,25,26] and responsible for the transport of fatty acyl-CoAs into peroxisomes[4–7].

Although ABCD1-3 all transport fatty acyl-CoAs, they differ from each other in substrate specificity. ABCD1 prefers to transport saturated and monounsaturated VLCFA-CoAs, such as C22:0-CoA[4,5], C24:0-CoA, C26:0-CoA[4,27] and C26:1-CoA[28]. Despite previous reports indicated that ABCD2 might transport polyunsaturated fatty acyl-CoAs in yeast[27] and mouse[29], the bona fide substrate of human ABCD2, which shares a sequence identity of 62% with ABCD1, remains unclear. In contrast, ABCD3, which is 39% sequence-identical to ABCD1, specifically transports CoA esters of dicarboxylic acid, branched-chain fatty acid, and the bile acid intermediates of di- and tri-hydroxycholestanoic acid[6,30].

Here, we report the structures of ABCD1 determined by single-particle cryogenic electron microscopy (cryo-EM) in three distinct states: the apo form at 3.5 Å, the substrate- and ATP-bound structures at 3.6 and 2.8 Å, respectively. Notably, a chimeric ABCD1 at high expression level, with the N-terminal 63 residues of human ABCD1 replaced by the 65 residues N-terminal segment of *Caenorhabditis elegans* homolog, was applied to structure determination and all activity assays, despite the N-terminal segment is not traceable in all three structures, as well as other recently reported ABCD1 structures[31,32]. Structural analysis combined with biochemical assays delineates snapshots of the transport cycle driven by ABCD1, and provides structural insights into the substrate specificity of fatty acid transporters. Moreover, these structures enable us to precisely map the pathogenic variants in *ABCD1* gene, and interpret the molecular basis of pathogenesis of X-ALD.

## Results

### Biochemical characterization and structure determination of ABCD1.
We initially overexpressed the full-length human ABCD1 (termed hABCD1 for short and hereafter) in HEK293F cells, but failed in purifying sufficient amount of recombinant proteins for further study. In contrast, we obtained a much higher yield of *C. elegans* PMP-4, a 52% sequence-identical homolog of hABCD1. Sequence alignment indicated that PMP-4 differs from hABCD1 mainly in the N-terminal segment (Supplementary Fig. 1a), which was proposed to be involved in the subcellular location[33–35]. Therefore, we constructed a chimeric ABCD1 (termed chABCD1) with the N-terminal 63 residues of hABCD1 replaced by the corresponding N-terminal 65 residues of PMP-4. The expression level of chABCD1 is ~3.5 folds to that of hABCD1 (Supplementary Fig. 1b), which enabled us to purify the homogenous chABCD1 in detergent micelles (Supplementary Fig. 1c). Thus, chABCD1 was applied to further activity assays and structure determination.

The ATPase activity assays showed that the activity of chABCD1 could be stimulated upon addition of various VLCFA-CoAs (C22:0-, C24:0- and C26:0-CoA), but not acetyl-CoA (Fig. 1a, b and Supplementary Fig. 1d). Moreover, C22:0-, C24:0- and C26:0-CoA displayed a similar $EC_{50}$ (half maximal effective concentration) value of ~2 μM and $V_{max}$ of ~200 mol Pi min$^{-1}$ mol$^{-1}$ protein (Fig. 1b), suggesting a similar in vitro specificity of ABCD1 toward these VLCFA substrates. Notably, the activity assays were performed in the optimized detergent combinations, namely 10:1 (w/w) of lauryl maltose neopentyl glycol (LMNG) and cholesteryl hemisuccinate (CHS).

To optimize the samples for the single-particle cryo-EM, we found chABCD1 behaves the best in digitonin, despite possessing a relatively lower ATPase activity (Supplementary Fig. 1c, e). Eventually, we solved three structures of chABCD1 (Fig. 1c): the apo form at 3.5 Å (Supplementary Fig. 2), C22:0-CoA bound form at 3.6 Å (Supplementary Fig. 3) and ATP-bound form at 2.8 Å (Supplementary Fig. 4). Notably, the fused N-terminal segment of PMP-4 is missing in all three structures most likely due to its high flexibility[31,32]. It indicated these structures indeed represent the core structure of human ABCD1; thus, the term ABCD1 was used for all structural descriptions.

### The apo-form structure of ABCD1.
The overall structure of ABCD1 exhibits a two-fold symmetric homodimer, each subunit of which contains a TMD and an NBD (Fig. 2a, b). The apo-form ABCD1 adopts an inward-facing conformation opening to cytosol, similar to the previously reported type-IV apo-form ABC homologs, including human ABCB6 and ABCB11, in addition to *Saccharomyces cerevisiae* Atm1 and *Campylobacter jejuni* PglK (Supplementary Fig. 5a). Each TMD of ABCD1 consists of six transmembrane helices (TMs), which are tightly packed against each other in the peroxisomal membrane leaflet, but split in the cytosolic membrane leaflet, and further extended into cytosol to form two diverged "wings". The helices TM4 and TM5 from one TMD are swapped to the opposite TMD, which is a typical feature of type-IV ABC transporters[18]. Two pairs of coupling helices from the TMDs are embedded in the grooves on the NBDs, coupling the conformational changes between TMDs and NBDs. At a membrane-exposed hydrophobic cleft between TM5 and TM6 of each subunit, there exists an extra density, the shape and size of which are reminiscent of a lipid molecule (Fig. 2a, b). It could be fitted with a phosphatidyl ethanolamine (PE) molecule

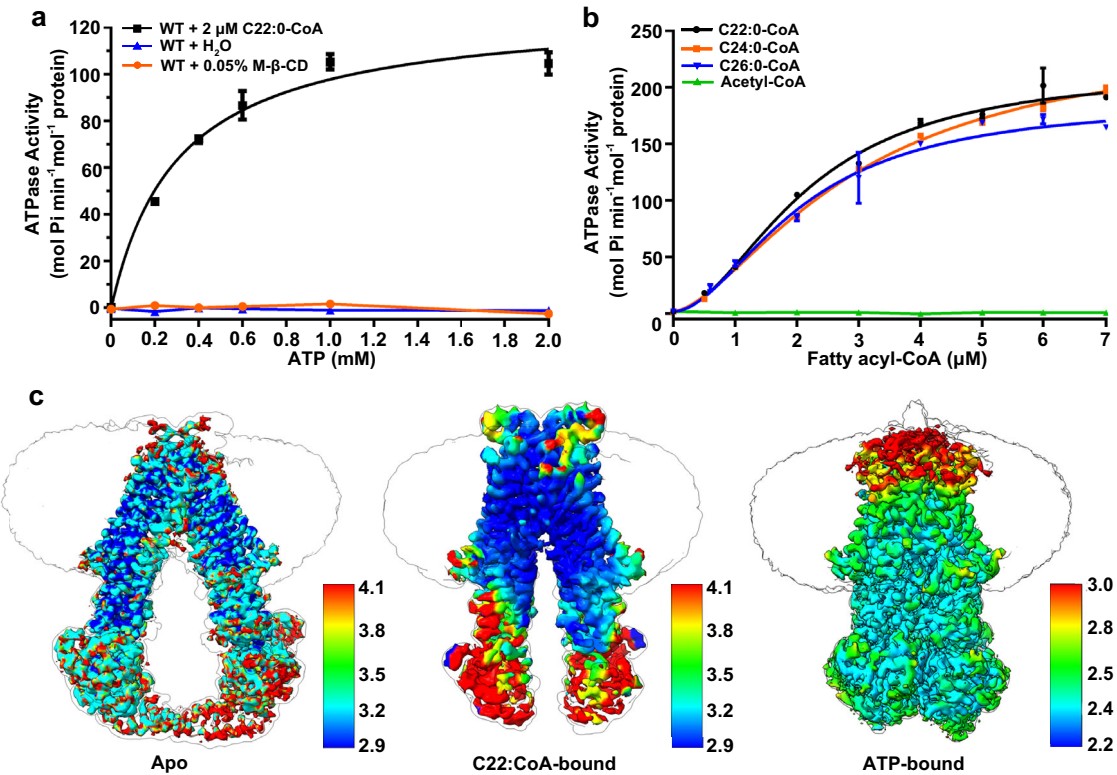

**Fig. 1 Substrate-stimulated ATPase activity assays and structure determination of ABCD1. a** ATPase activity of chABCD1 in the presence or absence of C22:0-CoA in detergent of LMNG + CHS. 0.05% methyl-β-cyclodextrin (M-β-CD) which functions as the solvent of substrate, was also tested as the control group. The data points were fitted with a Michaelis-Menten equation. **b** Substrate concentration-dependent ATPase activity of chABCD1 in detergent of LMNG + CHS and 2 mM ATP upon addition of varying fatty acyl-CoA. The data points were fitted with a Hill equation. All data points for **a** and **b** represent means of three independent measurements ($n = 3$) in detergent of LMNG and CHS. Error bars indicate standard deviation. Source data are provided as a Source Data file. **c** Refined cryo-EM map of three ABCD1 structures. The unsharpened maps are displayed as the outline to show the position of the detergent micelle. The cryo-EM map are colored by UCSF ChimeraX 1.2.5 according to the local resolution of the cryo-EM estimated by RELION 3.1 or cryoSPARC 3.1.

(Supplementary Fig. 5b), which might be extracted from the cell membrane during protein purification.

The NBDs possess a canonical NBD fold of ABC transporter, in addition to a couple of helices that form a crossover at the C-terminus (Fig. 2a). Sequence alignment revealed that this C-terminal helix is highly conserved in human ABCD1 and homologs (Supplementary Fig. 5c). Truncation of this helix led to a slightly increased $EC_{50}$ value toward C22:0-CoA (~4 μM) compared to the wild-type, but a sharply decreased $V_{max}$ ($17.5 \pm 1.6$ mol Pi min$^{-1}$ mol$^{-1}$ protein) of ATPase activity (Fig. 2c and Supplementary Fig. 5d, e). We thus speculated that the C-terminal helical crossover may facilitate the dimerization of two separated NBDs. In fact, a variant T693M on this C-terminal helix, which abolishes the function of ABCD1 without impacting the expression of ABCD1 in patients, was reported to be clinically associated with X-ALD[36]. It is in agreement with our ATPase activity assays of T693M mutant, which showed an almost non-detectable activity (Fig. 2c and Supplementary Fig. 5d, e). We suppose that the T693M mutation impairs the interactions between the two C-terminal helices, leading to the highly flexibility of the C-terminal helices, which somewhat interfere the dimerization of NBDs. However, ABCD1 with the truncated C-terminal helix still remains the capability of dimerization, thus retaining ~10% ATPase activity.

**The substrate-bound structure showed two symmetric C22:0-CoA molecules crosslinking the two TMDs.** Via addition of

0.5 mM C22:0-CoA, we obtained the complexed structure of substrate-bound ABCD1 (Fig. 3a, b). Superposition of this structure with the apo form yielded a root-mean-square-deviation (RMSD) of 9.2 Å over 1078 Cα atoms (Fig. 3c). Upon substrate binding, most TMs undergo a rigid-body shift toward the two-fold axis symmetry, which in consequence makes the two NBDs approach each other up to ~27 Å (Supplementary Fig. 6a). It results in an inward-facing conformation with a narrowed opening toward the cytosol, accompanying with the disappearance of the C-terminal helical crossover (Fig. 3c). We suppose that the conformational changes of NBDs upon substrate binding might break the interactions between the two C-terminal helices, which subsequently become highly flexible. Notably, the domain swapping helix TM4 located in the transport cavity unwinds into two helices (TM4a and TM4b). In addition, the segment of residues Glu363-Glu371 protruding to the peroxisomal matrix (Fig. 3a), which is unfolded in the apo form, gets folded into a short α-helix (termed α5a). Notably, a similar extracellular helix was also found in a type-IV ABC transporter, *C. jejuni* lipid-linked oligosaccharide flippase PglK[37].

In the TMDs of substrate-bound structure, we observed a pair of L-shaped densities, each of which could be well fitted with a molecule of C22:0-CoA (Fig. 3d and Supplementary Fig. 3e). The C22:0-CoA molecule traverses across two TMDs, with the free end of fatty acid almost extending to the interface between the TMD and peroxisomal matrix. The 3′-phospho-ADP moiety of CoA is buried in a hydrophilic pocket formed mainly by a series of positively-charged residues (Fig. 3e). The adenine ring is

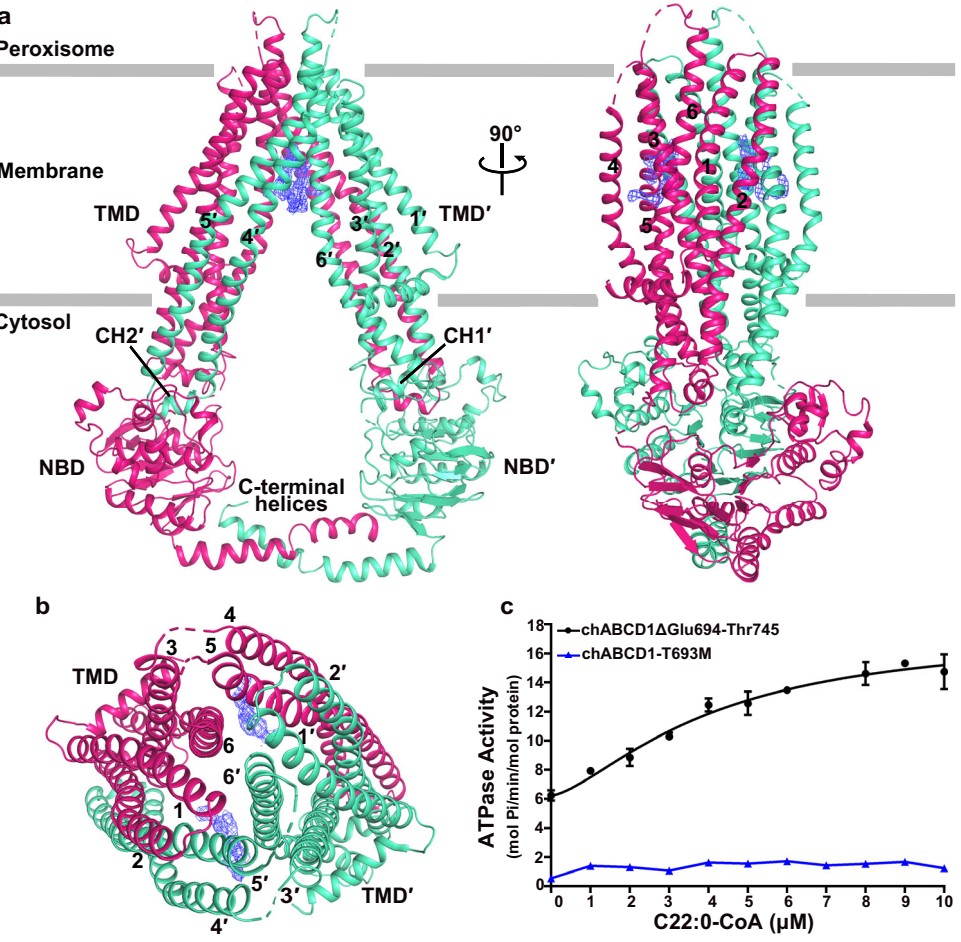

**Fig. 2 Overall structure of apo-form ABCD1. a** Cartoon representation of the apo-form ABCD1. Two subunits are colored in green and rose, respectively. Transmembrane helices (TMs), C-terminal helices and coupling helices (CH1′ and CH2′) are labeled. Two extra lipid-like densities are shown as the blue mesh. The peroxisome membrane is indicated as the gray lines. **b** Top view of ABCD1 (from the inside of peroxisome). The six TMs of each subunit are numbered. **c** The ATPase activity of the C-terminal helix truncated variant (ΔGlu694-Thr745) and T693M mutant in detergent of LMNG + CHS and 2 mM ATP upon addition of C22:0-CoA at various concentrations. Each data point is the average of three independent experiments ($n = 3$), and error bars represent the means ± SD. The data points were fitted with a Hill equation. Source data are provided as a Source Data file.

stabilized by Lys217 and Ser213 from TM3 via hydrogen bonds, in addition to the π-cation interaction between Arg401 from TM6 and the adenine ring. The 3′-phosphophate of ribose forms a salt bridge with Arg152 from TM2, whereas the diphosphate group interacts with Arg104 from TM1, Lys336′ and Tyr337′ from TM5′ via salt bridges and hydrogen bonds. The moiety of pantothenate and cysteamine, which links the fatty acyl chain and the 3′-phospho-ADP, extends from one TMD to the opposite TMD across the transport cavity (Fig. 3b, d), forming an inter-domain bridge in parallel to the membrane plane. The 22-carbon fatty acyl chain succeeding the moiety of pantothenate and cysteamine penetrates the opposite TMD, after turning 90° at the sixth carbon, and runs along the lateral of this TMD (Fig. 3f). The fatty acyl chain is embedded in a hydrophobic cleft formed by a cluster of hydrophobic residues, including Leu229′ and Ala233′ from TM3′, Val247′ and Val251′ from TM4′, Met346′, Val347′, Pro350′ and Ile351′ from TM5′, Ala384′, Ala388′, L392′, and Ala396′ from TM6′. In addition, an extra density at the peripheral of each TMD, which could be fitted well with a molecule of CHS (Fig. 3d), might provide an extra hydrophobic interface to align the fatty acyl chain (Fig. 3f).

Mutagenesis combined with activity assays indicated that the chABCD1 variants with single mutation of the substrate-binding residues, either those forming polar interactions with the CoA

portion (R104A, R152A, K217A, K336A, Y337F, and R401Q) or contributing hydrophobic interactions with the fatty acyl chain (A247W, G343V, P350W, and A395W) displayed significantly decreased substrate-simulated ATPase activity, compared to the wild-type (Fig. 3g and Supplementary Fig. 6b–d). The substitutions of A247, P350 and A395 to Trp might hinder the substrate binding, whereas the mutants R401Q and G343V have been reported to be associated with X-ALD[36]. In sum, the extensive polar and hydrophobic interactions ensure a highly specific binding pocket toward the CoA and fatty acyl chain portion of C22:0-CoA, respectively. Moreover, multiple-sequence alignment revealed that these substrate-binding residues are highly conserved in ABCD1 homologs (Supplementary Fig. 6e).

**The ATP-bound structure revealed a post-translocation state.** It was recently reported that the substrate can be translocated and released upon ATP binding to type-IV ABC transporters[38,39]. Thus, we constructed an E630Q mutant of chABCD1 to capture the ATP-bound conformation (Supplementary Fig. 7a). Notably, it indicated that the ATPase activity of E630Q mutant is not detectable, either in the presence or absence of 2 μM C22:0-CoA. (Supplementary Fig. 7b). Via adding ATP to the C22:0-CoA-bound E630Q mutant, we solved the ATP-bound form structure

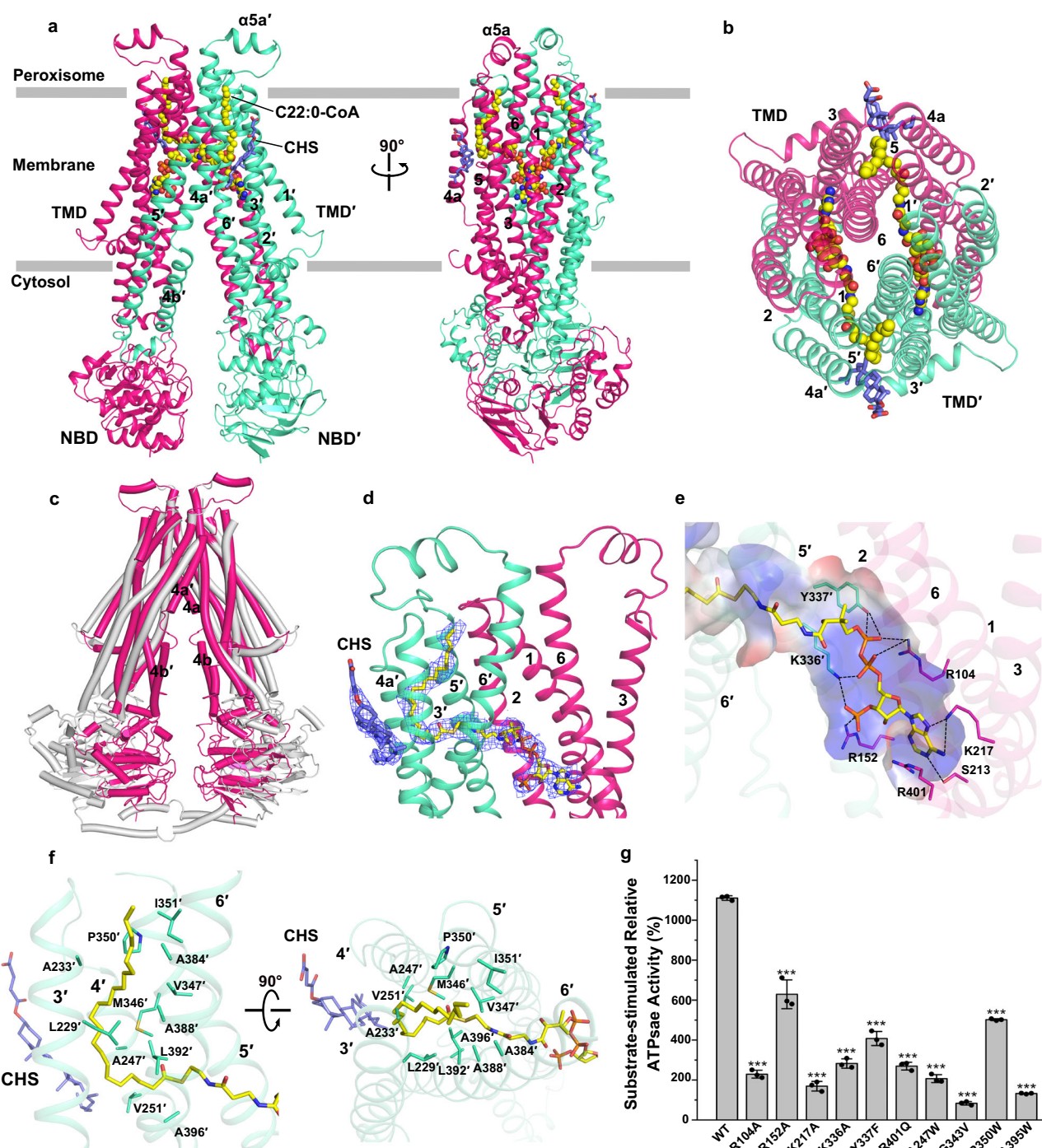

**Fig. 3 Structure of substrate-bound ABCD1 and the substrate binding pocket. a** Side and **b** Top view of the overall structure of C22:0-CoA-bound ABCD1. Two subunits of ABCD1 are colored in green and rose, respectively. The peroxisome membrane is indicated as the gray lines. The α-helices (termed α5a/5a') in the peroxisome matrix is labeled. The two C22:0-CoA molecules are shown as yellow spheres, and the two cholesteryl hemisuccinate (CHS) molecules are shown as blue sticks. **c** Superposition of the apo-form (gray) against C22:0-CoA-bound ABCD1 (pink). TM4a/4a' and TM4b/4b' are labeled. **d** The C22:0-CoA molecule binding to TMDs. The density map of C22:0 and CHS, shown as blue mesh, are contoured at 5σ. The CHS and C22:0-CoA are shown as blue and yellow sticks, respectively. **e** The binding pocket of the CoA portion of C22:0-CoA. The binding residues are shown as sticks, and hydrogen bonds (≤3.5 Å) and salt bridges (≤4.0 Å) are shown as black dotted lines. The electrostatic surface properties of the binding pocket are color-coded by electrostatic potential generated by PyMOL 2.5.2. **f** The binding pocket of fatty acyl chain of C22:0-CoA. The hydrophobic residues surrounding the fatty acyl chain within 4.5 Å are shown as sticks. The fatty acyl chain and CHS are shown as blue and yellow sticks, respectively. **g** Relative ATPase activities of chABCD1 and mutants in detergent of LMNG + CHS and 2 mM ATP upon addition of C22:0-CoA. The relative activity represents the substrate-stimulated activity of chABCD1 or its mutant that harboring a single mutation of residues at the substrate-binding pocket. Each data point is the average of three independent experiments ($n = 3$), and error bars represent the means ± SD. One-way analysis of variance (One-way ANOVA) is used for the comparison of statistical significance of mutants and wild-type. The p value of R104A, R152A, K217A, K336A, Y337F, R401Q, A247W, G343V, P350W and A395W is 0.00019, 0.00062, 0.00076, 0.00092, 0.00053, 0.00063, 0.00014, 0.000178, 0.00031 and 0.00012. The p values of <0.05, 0.01, and 0.001 are indicated with *, ** and ***. Source data are provided as a Source Data file.

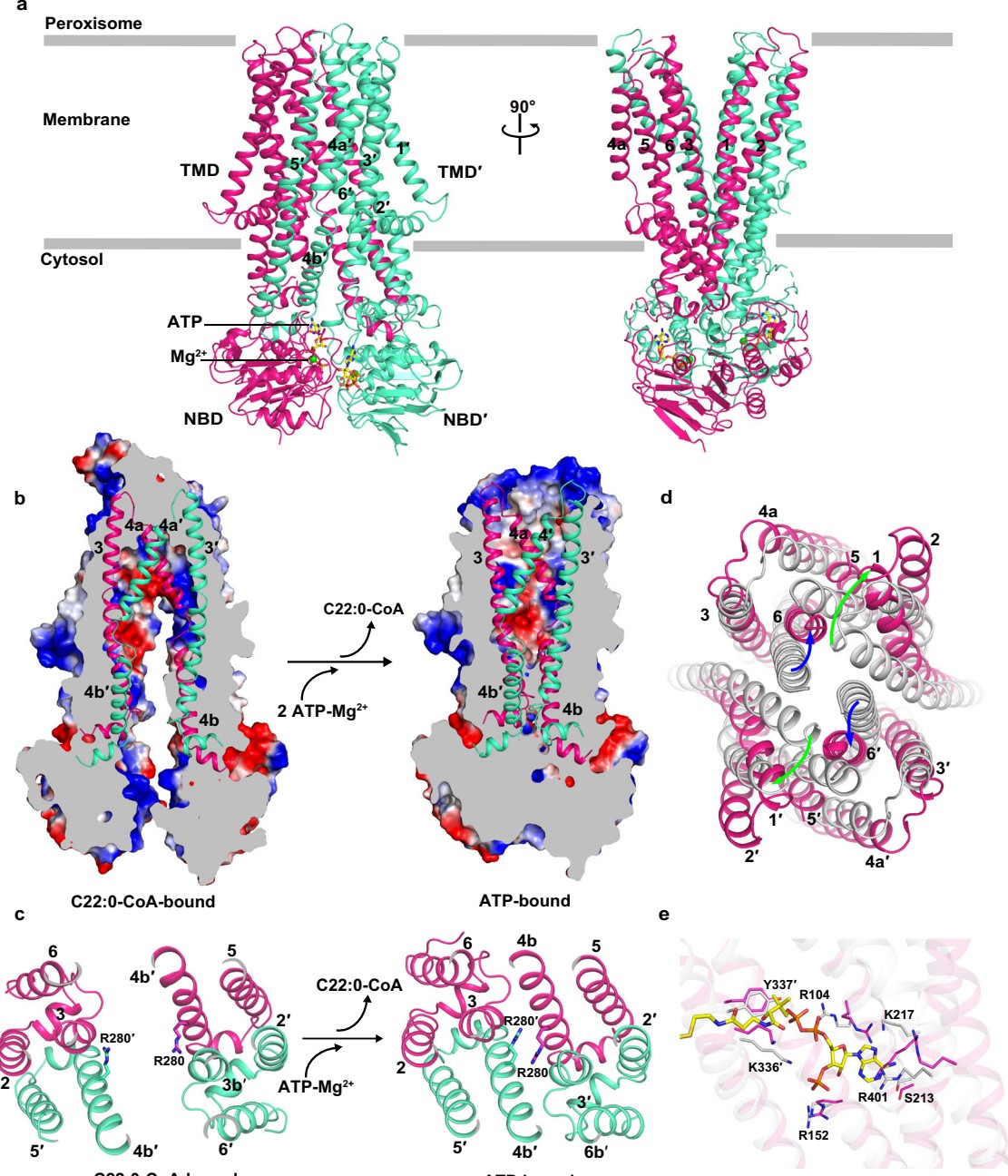

**Fig. 4 Structure of ATP-bound ABCD1 and comparison with the C22:0-CoA-bound structure. a** Cartoon representation of ATP-bound ABCD1. Two subunits of ABCD1 are colored in green and rose, respectively. ATP molecules are shown as yellow sticks, and Mg$^{2+}$ ions are displayed as green spheres. The number of TMs are labeled. Conformational changes upon ATP binding accompanied with substrate release, as shown by **b** cutaway representation of the electrostatic surface, and **c** top view of C22:0-CoA-bound-ABCD1 and ATP-bound-ABCD1. TM3/TM3' and TM4/TM4' are shown as cartoon. The electrostatic surface was generated by PyMOL 2.5.2. The residue Arg280 is shown as sticks. Superposition of (**d**) the TMDs and (**e**) the CoA portion binding pocket of C22:0-CoA-bound ABCD1 (gray) against the ATP-bound form (rose). TM helices are shown as cartoons and numbered. The shifts of TM1/TM1' and TM6/TM6' are indicated in green or blue arrows, respectively. The residues interacting with CoA portion of C22:0-CoA are displayed as sticks.

at a resolution of 2.8 Å (Fig. 4a and Supplementary Fig. 4), which resembles the ATP-bound ABCD4 with an RMSD of 1.8 Å over 706 Cα (Supplementary Fig. 7c). The structure showed excellent side-chain densities for most segments of the protein, except for the C-terminal helical crossover and helix α5a that were respectively assigned in the apo and substrate-bound forms. As expected, despite 0.5 mM C22:0-CoA was added prior to the addition of 20 mM ATP-Mg$^{2+}$, the substrate is absent in the

ATP-bound structure, which most likely represents a state succeeding substrate release upon ATP binding.

In this structure, two unambiguous densities, which are symmetrically sandwiched between the two NBDs, could be fitted with two ATP-Mg$^{2+}$ molecules (Supplementary Fig. 7d). The two ATP molecules bind to the cleft between the Walker A motif of one NBD and the ABC signature motif of the opposite NBD. The two NBDs are dimerized and adopt a typical "head-to-

tail" configuration[18]. In consequence, the conformational changes in NBDs are transferred to TMDs via two pairs of coupling helices, and eventually make the ATP-bound ABCD1 exhibiting an outward-facing conformation (Fig. 4a).

Upon ATP binding, the helices TM3 and TM4b from one TMD, together with the corresponding TM3′ and TM4b′ from the opposite TMD, tilt toward the central axis, forming a closed transport cavity at the cytosolic side (Fig. 4b). Particularly, the residues Arg280 from TM4b and Arg280′ from TM4b′ approach each other, and ultimately seal the transport cavity (Fig. 4c), which might serve as a cytosolic gate. Notably, in both substrate- and ATP-bound forms, Arg280 from TM4 is stabilized by Asp194′ and Glu199′ from TM3′ via salt bridges (Supplementary Fig. 7e). Seeing inside of the peroxisome, the helices TM1 and TM6 tilt outwards upon ATP binding, forming an enlarged opening toward the peroxisomal matrix (Fig. 4d). The hydrophobic pocket of the fatty acyl chain, which formed by TM3, TM4, TM5 and TM6, is destroyed due to the pronounced conformational changes of TM5 and TM6. Meanwhile, most of the CoA-binding residues are misaligned, especially Lys217 that occupies the position of the adenine ring of the 3′-phospho-ADP (Fig. 4e). All together, the substrate-binding pocket is completely collapsed, resulting in exclusion of the substrate from an enlarged opening to the peroxisomal matrix.

## Discussion
Our present structures provided the molecular insights into the etiopathology of pathogenic variants in the *ABCD1* gene. To date, more than 900 pathogenic variants in the ABCD1 gene have been reported in the X-ALD Database (https://adrenoleukodystrophy.info/), in which 231 (involved in 152 residues) are X-ALD associated missense mutations[14]. By mapping these 152 sites on the C22:0-CoA-bound ABCD1 structure (Supplementary Fig. 8 and Supplementary Table 2), we found that up to 67 sites are located on the NBDs, including 15 residues directly participating in ATP-$Mg^{2+}$ binding, in addition to Thr693 at the C-terminal helix. Besides, 35 pathogenic variants are clustered around the substrate-binding pocket, including Arg401 and Gly343. In total 43 pathogenic variants may affect the conformational coupling between TMDs and NBDs; for instance, pathogenic variant of Gly266, which is localized at the kink of TM4, might alter the unfolding-and-refolding capacity of TM4 in the transport cycle. Moreover, seven pathogenic variants at the N-terminus have been reported, including Arg74, Pro84, Leu88, Glu90, Ala95, His97 and Ser98, which might be important for recognizing PEX19 that targets ABCD1 to the peroxisomal membrane[40].

Besides our present structures, a couple of human ABCD1 structures have been recently reported[31,32,41,42], including 3 ATP-bound, 2 apo-form and 2 substrate-bound forms available in total[31,32]. All three ATP-bound structures are almost identical (Supplementary Fig. 9a), except for a bending TM6 of ABCD1 in the nanodisc structure (PDB ID: 7SHM). Compared to our apo-form structure, the apo-form structure in the nanodisc at 4.4 Å (PDB ID: 7RRA) also adopts an inward-facing conformation, but slightly narrower opening (Supplementary Fig. 9b). Superposition of the oleoyl-CoA-bound ABCD1 (PDB ID: 7HSN) against our apo-form (Supplementary Fig. 9c) and substrate-bound structures (Supplementary Fig. 9d) revealed an RMSD of 1.2 Å over 1039 Cα and of 9.2 Å over 1152 Cα, respectively. This discrepancy might come from the different substrate-binding stage, or the unfavored substrate oleoyl-CoA, which was reported to be oxidized in mammalian mitochondria[43,44]. In addition, the ABCD1 structure predicted by AlphaFold2[45–47] showed a conformation similar to our substrate-bound structure (Supplementary Fig. 9e, f), with an RMSD of 2.3 Å over 532 Cα.

To date, there are only four structures of lipid-bound ABC transporters, namely *Escherichia coli* lipopolysaccharide flippase MsbA[48,49], human phospholipid transporter ABCB4[50] and ABCA4[51,52], in addition to *E. coli* phospholipid transport MlaFEDB[53]. For all these structures, only one substrate molecule is bound to the transport cavity between two TMDs. In contrast, our structure of C22:0-CoA-bound ABCD1 revealed a distinct substrate binding pattern: two L-shaped C22:0-CoA molecules symmetrically cross the two TMDs. This binding pocket across two subunits provides an extended binding site long enough to accommodate VLCFA-CoA; in turn, the substrate serves as a crosslinker that further stabilizes the homodimer of ABCD1 and enhances the inter-subunit crosstalk besides the extensive domain swapping. In fact, the substrate-stimulated ATPase activities over C22:0-CoA at increasing concentrations revealed a Hill coefficient of ~1.9 (Fig. 1b), indicating a significant cooperativity between the two subunits of the half transporter ABCD1. Moreover, this cooperativity did not come from the two ATP-binding sites, as the ATPase activity over varying ATP concentrations, in the presence of a fixed concentration of C22:0-CoA, revealed a Michaelis-Menten curve (Fig. 1a).

The present structures of ABCD1 at three different states combined with activity assays enabled us to propose a model of ABCD1-mediated VLCFA-CoA transport (Fig. 5). At the resting state, ABCD1 adopts an inward-facing conformation, with the two NBDs properly distanced controlled by the C-terminal helical crossover. Due to the induced fit upon the cooperative binding of substrates, the two TMDs shift toward each other, forming a narrowed inward-facing transport cavity, accompanying with dissociation of the C-terminal helical crossover. Thus the two NBDs approach each other and get dimerized upon ATP binding; in consequence, ABCD1 adopts an outward-facing conformation, which leads to a collapsed substrate-binding pocket and eventually enables the release of substrate to the peroxisomal matrix. Finally, the phosphate release succeeding to hydrolysis of ATP triggers conformational changes and resets ABCD1 to the resting state for another transport cycle.

## Methods
**Protein expression and purification.** The codon-optimized full-length human *ABCD1* gene encoding the ABCD1 protein (Uniprot ID: P33897) and *Caenorhabditis elegans pmp*-4 encoding the PMP4 protein (Unirprot ID: O45730) were synthesized by GENEWIZ Company. The chimeric *ABCD1* gene, in which the N-terminal region of 189-bp was replaced by the N-terminal region of 195-bp from *pmp-4*, was generated by overlap PCR. The full-length human *ABCD1* gene and chimeric ABCD1 (termed chABCD1 hereafter) was cloned into a modified pCAG vector with an N-terminal FLAG tag (DYKDDDDK) using a ClonExpress® II One Step Cloning Kit (C113-02, Vazyme Biotech co., Ltd). Site-directed mutagenesis was performed using a standard two-step PCR and verified by DNA sequencing (Shenggon Biotech, shanghai).

For protein expression, the 800 ml HEK293F cells (R79007, Invitrogen) were cultured in SMM 293T-II medium (M293TII, Sino Biological Inc.) at 37 °C, under 5% $CO_2$ in a shaker. When the cell density reached ~2.5 × 10⁶ cells per ml, ~2 mg plasmids with 4 mg linear polyethylenimines (23966-2, Polysciences Co., Inc) were pre-incubated in 45 ml fresh medium for 15 min, which was then added to the cell culture with the supplement of another 150 ml fresh medium, followed by 30-min static incubation. The transfected cells were grown at 37 °C for 12 h. Afterwards, sodium butyrate (S102954-500g, Aladdin) was added with a final concentration of 10 mM, followed by cultivation at 30 °C for additional 48 h before harvest. After centrifugation at 4000 × *g* for 5 min, the cell pellets were resuspended in the lysis buffer containing 25 mM Tris-HCl pH 7.5, 150 mM NaCl, 20% glycerol (v/v), 1 mM dithiothreitol (DTT) and 1 × protease inhibitor cocktail (C0001, TargetMol). The suspension was frozen quickly in liquid nitrogen and stored at −80 °C for further use.

The membrane proteins were extracted from the cells with 1% (w/v) LMNG (NG310, Anatrace), 0.1% (w/v) CHS (C6013-25, Anatrace) and rotated gently at 4 °C for 2 h. After centrifugation at 45,000 rpm for 45 min (Beckman, Type 70 Ti), the supernatant was collected and applied to anti-FLAG M2 affinity gel (Sigma-Aldrich) at 4 °C for 1 h. Then the resin was rinsed with buffer A containing 25 mM Tris-HCl pH 7.5, 150 mM NaCl, 10% glycerol (v/v), 1 mM DTT, 0.06% digitonin (w/v) (BID3301, Apollo Scientific). The protein was eluted with buffer B containing 25 mM Tris-HCl pH 7.5, 150 mM NaCl, 5% glycerol (v/v), 1 mM DTT, 0.06%

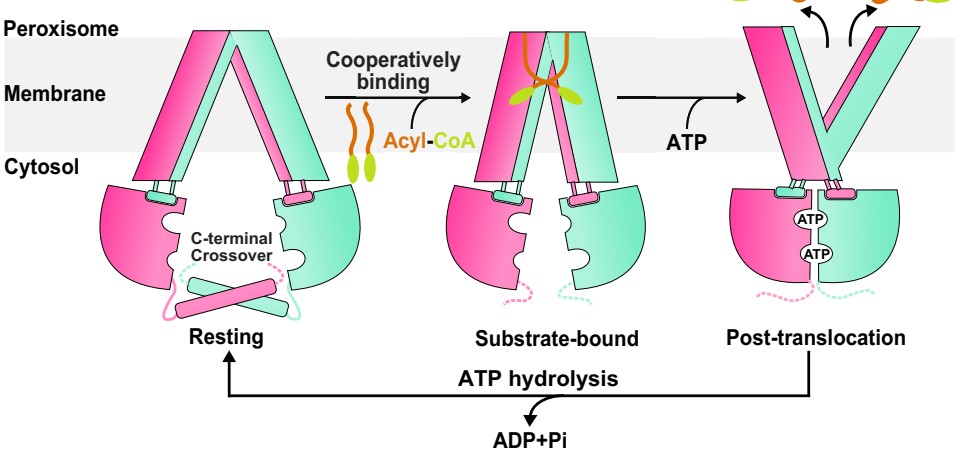

**Fig. 5 A working model for the ABCD1-mediated fatty acyl-CoA translocation.** Schematic illustration of the transport cycle of ABCD1, inferred from the three resolved structures. The apo-form ABCD1 adopts an inward-facing conformation, with the two NBDs properly separated via the C-terminal helical crossover (resting state). Cooperative substrate binding drags two TMDs approaching each other, resulting in a narrowed inward-facing transport cavity (substrate-bound state). Upon ATP binding, the NBDs approach each other and dimerize, making ABCD1 in an outward-facing conformation, and facilitating the substrate release into the peroxisome (post-translocation state). Finally, the hydrolysis of ATP resets ABCD1 to the resting state, and ready for another transport cycle. The CoA portion and fatty acyl chain of fatty acyl-CoA are shown as green ovals and brown curved lines, respectively.

digitonin (w/v) supplemented with 200 µg/ml FLAG peptide. The protein eluent was concentrated by a 100-kDa cut-off Centricon (Millipore) and further purified by size-exclusion chromatography using a Superdex 200 Increase 10/300 or Superose 6 SEC column (Cytiva) or equilibrated with wash buffer C containing 25 mM Tris-HCl pH 7.5, 150 mM NaCl, 1 mM DTT, 0.06% digitonin (w/v). Peak fractions were pooled and concentrated for further biochemical studies or cryo-EM experiments.

The protein used for ATPase activity assays were expressed and purified in the same way except that the detergent was for buffer A, B, C were substituted by 0.005% LMNG and 0.0005% CHS.

All the mutant proteins using in this project were expressed and purified in the same way as the wild-type protein.

**ATPase activity assay**. The ATPase activities of chABCD1 and mutants were measured using the ATPase Colorimetric Assay Kit (Innova Biosciences) in 96-well plates at $OD_{630\,nm}$. All compounds used in the ATPase activity assays, including acetyl-CoA (trilithium salt) (A2181, Sigma-Aldrich), C22:0-CoA (behenoyl coenzyme A, ammonium salt) (870722P, Sigma-Aldrich), C24:0-CoA (lignoceroyl coenzyme A, ammonium salt) (870724P, Sigma-Aldrich), C26:0-CoA (hexacosanoyl coenzyme A, ammonium salt) (870726P, Sigma-Aldrich), were purchased from Sigma-Aldrich and were dissolved in 5% (w/v) methyl-β-cyclodextrin (M-β-CD), (H0388-25MG, Sigma-Aldrich).

To measure the ATPase activities of chABCD1 against different substrates or varying ATP concentrations, protein at a final concentration of 0.03 µM was added to the reaction buffer containing 20 mM Tris-HCl, pH 7.5, 50 mM KCl, 1 mM DTT, 0.005% (w/v) LMNG/0.0005% (w/v) CHS, 2 mM $MgCl_2$. Then, each substrate was diluted into different concentrations and added into the reaction mixture and the final concentration of the methyl-β-cyclodextrin is 0.05% (w/v). The mixture was incubated on the ice for 10 min, then ATP was supplemented in the solution with a final concentration of 2 mM (for ATPase activity assay of chABCD1 against different substrates) or a varying ATP concentration to start the reaction at 37 °C for 20 min. The amount of released Pi was quantitatively measured and statistical analysis was performed using Origin 2021b (Academic).

In the substrate-stimulated relative ATPase activity assays, all the procedures are the same as above, except that the final protein concentration is 0.1 µM and the reaction time is 45 min. The relative activities were calculated by normalizing the ATPase activity in the presence of 5 µM C22:0-CoA relative to that in the absent of C22:0-CoA.

**Western blot analysis**. The whole cell lysate samples were run on BeyoGel™ Plus Precast PAGE Gel (4 to 10%) for Tris-Gly System (P0465S, Beyotime) for 110 min at 110 V. Gels were transferred to PVDF membranes with the Trans-Blot Turbo Transfer System (Bio-Rad). The membranes were blocked by 5% (w/v) non-fat milk (A600669-0250, Sangon Biotech) at 4 °C overnight, before sliced into stripes according to a pre-stained ColorMixed Protein Marker (PR1920, Solarbio). The sliced stripes were incubated with primary antibody for 1 h at ambient temperature. β-actin were used to be a reference protein for Western blot in this project. Then the membranes were washed five times with TBST buffer before incubation with secondary antibody for 30 min at ambient temperature. The membranes were washed three times and imaged via ImageQuant LAS4000 (Cytiva). For semi-

quantitative analysis, band intensities were measured by densitometry using the software ImageJ 1.38X. Primary antibodies used in this study include: FLAG tag mouse monoclonal antibody (1:20,000) (66008-3-Ig, Proteintech) and β-actin mouse monoclonal antibody (1:5000) (66009-1-Ig, Proteintech). Secondary antibody used in this study is peroxidase-conjugated Affinipure Goat Anti-Mouse IgG (H + L) (1:10,000) (SA00001-1, Proteintech).

**The thermal stability assay**. The thermal stability assays were performed using a modified protocol based on a previous described method for cellular thermal shift assay[54]. Proteins purified in LMNG+/CHS were concentrated to ~0.2 mg/ml and aliquots (15 µl) were exposed to various temperatures from 37 to 61 °C using a thermocycler PCR machine (Bio-Rad) for 5 min. Then, protein samples from each thermal point were centrifuged at $14,000 \times g$ for 10 min to remove the precipitated proteins. Proteins in the supernatant were visualized by Coomassie-blue stained SDS-PAGE and quantified by densitometry using the software ImageJ 1.38X. Sample quantities in each group were normalized by the sample at 37 °C and all data points in each group were fit with a Boltzmann equation to calculate melting temperature ($T_m$).

**Cryo-EM sample preparation and data collection**. To prepare the apo-form ABCD1 sample, aliquots of 3.5 µl purified chABCD1 at a concentration of ~5 mg/ml was applied to glow-discharged QUANTIFOIL (R1.2/1.3, 300 mesh, holey carbon films) Cu grids. The grids were blotted with filter paper for 4.0 s and 0 blotting force. Then, the grids were plunged into liquid ethane cooled with liquid nitrogen using a Vitrobot Mark IV (FEI) under 100% humidity at 8 °C. Two datasets with a total of 7495 micrograph stacks were automatically collected with SerialEM[55] on a Titan Krios microscope at 300 kV equipped with a K3 Summit direct electron detector (Gatan) at a nominal magnification of ×22,500 with defocus values from −2.0 to −1.5 µm. Each movie stack that contains 32 frames was exposed in a super-resolution mode, with a total dose of 60 e−/Å².

To prepare C22:0-CoA-bound ABCD1 complex sample, the purified chABCD1 concentrated to ~5 mg/ml was mix with 0.5 mM C22:0 CoA and incubated for 30 min on the ice. After that, aliquots of 3.5 µl protein mixture were applied to glow-discharged QUANTIFOIL (R1.2/1.3, 300 mesh, holey carbon films) Cu grids. The grids were blotted with filter paper with a 4 s blotting time and −1 blotting force. Then the grids plunged into liquid ethane cooled with liquid nitrogen using a Vitrobot Mark IV (FEI) under 100% humidity at 8 °C. Three datasets with a total of 11,617 micrograph stacks were collected in the same manner as apo-form ABCD1. These stacks were motion corrected with dose weighting by MotionCor2[56] with a binning factor of 2, resulting in a pixel size of 1.07 Å. The defocus values were estimated using CTFFIND4[57].

To prepare ATP-bound ABCD1 complex sample, the purified chABCD1-E630Q concentrated to ~8 mg/ml was mix 0.5 mM C22:0 CoA and incubated for 30 min on the ice. Then the protein mixture were mixed with 20 mM ATP and 20 mM $MgCl_2$ and incubated on the ice for 30 min. After that, aliquots of 3.5 µl protein mixture were applied to glow-discharged QUANTIFOIL (R1.2/1.3, 300 mesh, holey carbon films) Cu grids. The grids were blotted with filter paper with a 5 s blotting time and 0 blotting force. Then the grids were plunged into liquid ethane cooled with liquid nitrogen using a Vitrobot Mark IV (FEI) under 100% humidity at 8 °C. In total, 4302 micrograph stacks were collected with EPU 2 software[58] on a Titan Krios microscope at 300 kV equipped with a K3 detector

(Gatan) and a GIF Quantum energy filter (Gatan), at a nominal magnification of ×81,000 with defocus values from −2.0 to −1.8 μm. For these stacks, motion correction and dose weighting were performed with patch motion correction with a Fourier cropping factor of 0.5, resulting in a pixel size of 1.07. Meanwhile, the defocus values were estimated using Patch CTF estimation[59].

**Cryo-EM data processing**. For the apo-form ABCD1 datasets, 2,194,963 and 1,899,981 particles were automatically picked from two datasets, using RELION 3.1[60], respectively. After 2D classification, 1,174,746 particles from dataset 1 and 900,138 particles from dataset 2 were selected and subjected to 3D classifications with global search. Then we merged all good particles from two datasets after several rounds of 3D classification applying C1 symmetry. Finally, 360,311 particles were selected for 3D auto-refinement with an adapted mask and yielded a reconstruction with an overall resolution of 3.5 Å, followed by CTF refinement and Bayesian Polishing.

For C22:0-CoA-bound ABCD1 datasets, 1,860,455 particles, 2,583,043 particles and 2,500,479 particles from three datasets were respectively picked and subjected to 2D classification and initial global search 3D classification with a C1 symmetry. Then C2 symmetry was imposed for further global 3D classification. In total, 467,598 particles were selected from three datasets and imported into cryoSPARC 3.1[59] for ab initio reconstruction and heterogeneous refinement. Ultimately, 336,741 particles were selected for further refinement and yielded a reconstruction map at 3.6 Å with C2 symmetry.

The procedures for ATP-bound ABCD1 were performed on the cryo-SPARC 3.1[59]. Totally 1,878,754 particles were automatically picked from 4302 micrographs and subjected to 2D classification. Afterwards, 972,858 particles were selected and used to ab initio reconstruction and heterogeneous refinement with a C1 symmetry. Then 550,864 particles were selected and applied to four rounds of ab initio reconstruction and heterogeneous refinement with C2 symmetry, yielding a reconstruction map at a resolution of 2.8 Å.

All resolutions of the cryo-EM maps were estimated using the gold-standard Fourier shell correlation 0.143 criterion[61].

**Model building and refinement**. Since density for the segment residues from PMP-4 missed in all the three reconstruction maps, we numbered all the residues in these three structures corresponding to the sequence number of human ABCD1. The model of apo-form ABCD1 was manually rebuilt in COOT 0.8.9.2[62] and refined using Real-space refinement in PHENIX 1.18.2 with secondary structure and geometry restraints. Two extra densities between TM5 and TM6 can be observed in each TMD, each of which was fitted by an 18:0 Lyso PE molecule. In the structure of C22:0-CoA-bound ABCD1, two C22:0-CoA molecules and two CHS molecules were manually built in the map. Due to their relatively poor density, the NBDs was manually fitted into the map using the NBD structure from ATP-bound ABCD1, and was automatically refined in PHENIX 1.18.2[63]. For the ATP-bound ABCD1, an outward-facing model of ABCD1 was generated by the SWISS-MODEL server[64], using the cryo-EM structure of human ABCD4 (PDB code 6jbj) as the reference model. Residues 64-345, 383-435 for TMDs and 461-685 for NBDs were built in the map, and two prominent densities between two NBDs allowed us to fit two ATP-Mg$^{2+}$.

All structures were validated by PHENIX 1.18.2[63] and MolProbity 4.02[65]. The model refinement and validation statistics were summarized in Table S1. The UCSF ChimeraX 1.2.5[66] and PyMOL 2.5.2 (https://pymol.org) were used for preparing the structural figures. Protein sequences were aligned using Multalin (http://multalin.toulouse.inra.fr/multalin/) and the sequence-alignment figures figures were generated by ESPript 3 server (https://espript.ibcp.fr/).

**Reporting summary**. Further information on research design is available in the Nature Research Reporting Summary linked to this article.

## Data availability
The cryo-EM density maps of three structures have been deposited at the Electron Microscopy Data Bank under accession codes: EMD-32152 for apo-form ABCD1, EMD32224 for C22:0-CoA-bound ABCD1 and EMD-32171 for ATP-bound ABCD1 and coordinates have been deposited at PDB under accession codes: 7VWC for apo-form ABCD1, 7VZB for C22:0-CoA-bound ABCD1 and 7VX8 for ATP-bound ABCD1. Source data are provided with this paper.

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

## Acknowledgements
We thank Dr. Yongxiang Gao at the Center for Integrative Imaging, University of Science and Technology of China during cryo-EM image acquisition. We thank Xiaojun Huang and Xujing Li at the Center for Biological Imaging at the Institute of Biophysics (IBP), Chinese Academy of Sciences for technical support on cryo-EM data collection. This work was supported by the Ministry of Science and Technology of China (2020YFA0509302), National Natural Science Foundation of China (32071206) and The Strategic Priority Research Program of the Chinese Academy of Sciences (XDB37020202).

## Author contributions
Y.C. and W.-T.H. conceived the project and planned the experiments. Z.-P.C., D.X., Y.-X.M., Y.L. and M.-T.C. expressed and purified proteins used in this project. Z.-P.C. and L.W. performed cryo-EM sample preparation, data collection, structure determination and model refinement. Z.-P.C. performed functional assays. W.-T.H., Z.-P.C., Y.C. and C.-Z.Z. wrote the manuscript. All authors read and edited the paper.

## Competing interests
The authors declare no competing interests.
