## [Peer Review File · Nature Communications]

Structural basis of substrate recognition and translocation by human very long-chain fatty acid transporter ABCD1REVIEWER COMMENTS

Reviewer #1 (Remarks to the Author):

ABCD1 is a type IV ATP-binding cassette (ABC) transporter that catalyzes the translocation of long-chain and very-long-chain fatty acids (VLCFAs) into peroxisomes. Dysfunctional ABCD1 is the root cause of adrenoleukodystrophy (ALD), a rare disease affecting the nervous system and adrenal glands.

Chen et al. present three cryo-EM reconstructions of detergent-solubilized homodimeric ABCD1 (chimera of human ABCD1 containing N-terminal segment of *C. elegans* ABCD1) in the apo, substrate-bound, and ATP-bound states at overall resolutions of 3.5, 3.6, and 2.8 Å, respectively. First, the authors demonstrate that their recombinantly expressed chimera can catalyze ATP hydrolysis and that VLCFA-CoAs stimulate the ATPase activity of ABCD1. The apo structure exhibits the typical inward-facing conformation of type IV systems. An interesting feature of the apo state are dimerized C-terminal helices whose truncation leads to a significantly lower V_{max} , suggesting that these helices facilitate NBD dimerization and ATP hydrolysis, similar to the TAP-related heterodimeric transporter TmrAB (add ref. doi:10.1073/pnas.1620009114). This notion is corroborated by a disease-associated mutation in the helices. Interestingly, the substrate-bound structure shows that binding of C22:0-CoA induces a rigid-body movement of the TMDs, with the NBDs moving closer together and dissolving the interaction of the C-terminal helices. These conformational changes are accompanied by the formation of a short helix on the peroxisomal side of the transporter, similar to a helix in PgIK (ref. 34). Chen et al. identify two molecules of C22:0-CoA in their map, and each molecule traverses across the two TMDs. The binding sites observed in the cryo-EM maps are confirmed by ATPase measurements with mutants harboring changes in the substrate-coordinating residues. Interestingly, two of the substrate-coordinating residues have also been described to be linked to ALD. Finally, the ATP-bound state adopts the typical outward-facing conformation and, similar to other type IV ABC transporters, most likely represents the stage after substrate release, as the substrate-binding sites are destroyed in this conformation.

The work by Chen et al. defines the structural basis of substrate binding and specificity of ABCD1 and outlines the conformational changes associated with different stages of the translocation cycle. Moreover, the authors map disease-causing mutations onto their ABCD1 structures and attempt to rationalize their pathogenic effects.

This work delivers significant new insights into the translocation mechanism of an important subfamily of ABC transporters, which warrant publication in Nature Communications. Although reasonably well written, it should be proofread again because it contains quite a few syntax/grammar/spelling mistakes throughout the text. The authors should also address the following points before publication:

Major points:

1) Fig. 3e: Some of the interaction distances (dashed lines) appear to be quite large. What cutoff did the authors choose to define interactions (hydrogen bonds and salt bridges)? Some dashed lines also do not seem to depict the interactions with the shortest distance (i.e., the strongest interactions), e.g., for R104.

2) Some aspects of the apo and substrate-bound models should be improved: (i) apo: rotamers (and possibly Ramachandran outliers), (ii) substrate-bound: bond angles, clashscore, and rotamers.

3) The authors should mention the final size of the purified construct used for cryo-EM analysis. What will happen to the chimeric N-terminus? Is it processed further? The mature N-terminus of the construct should be determined by Edman sequencing.

4) For the general readership, it would be helpful to mention in the introduction that ABCD1-4 are classified as type IV ABC transporters based on their structure and transmembrane fold (ref. doi:10.1146/annurev-biochem-011520-105201; doi:10.1002/1873-3468-13936).

5) Lines 207-210: The authors might consider removing the term “type II ABC exporter”, as they correctly point out that transporters with homologous TMDs include importers; thus, “type IV and type V ABC transporters” might indeed be the more appropriate term. => see also lines 327/328.

6) Line 160: The proof that substrates are translocated and released upon ATP binding have experimentally been demonstrated only recently for type IV transporter by single turnover experiments (consider ref. doi:10.7554/eLife.55943; doi:10.7554/eLife.67732 instead of citing a review).

7) Line 162: The authors might consider that an E630Q does not abolish the ATP hydrolysis. If the authors were to use a more sensitive assay, a slowdown of ATP turnover would be observed (again, see ref. doi:10.7554/eLife.55943).

8) In the abstract, the authors state that C22:0-CoA binds cooperatively to the transmembrane domains. However, no experimental evidence is provided for this statement. The Hill coefficient of 1.9 in substrate-stimulated ATPase activity likely reflects ATP binding and hydrolysis at the two nucleotide-binding sites.

Minor points:

- 1) In the abstract: Please provide the full name of fatty acyl-coenzyme A (C22:00-CoA).
- 2) Line 21: Consider removing “clearly” as illustrated snapshots should already be defined.
- 3) Line 31: “Cytoplasm” is the wrong term as it includes the interior of the cell except the nucleus, so organelles such as peroxisomes are part of the cytoplasm. “Cytosol” would be the correct term. Please check carefully.
- 4) Line 47, 49, 50, etc.: “ABCD1-4 or ABCD1-3” instead of “ABCD1~4 or ABCD1~3”.
- 5) Line 15, 61: “cryogenic electron microscopy” instead of “cryo-electron microscopy”.
- 6) Line 146: “might provide” instead of “might provides”.
- 7) Line 224: “approach each other” instead of “approach towards each other”.
- 8) Lines 226/227: It is phosphate release after ATP hydrolysis rather than ATP hydrolysis itself that resets the transporter to the inward-facing state (ref. doi:10.1038/s41586-019-1391-0; doi:10.7554/eLife.55943).
- 9) Line 500: “maps” instead of “map”

Reviewer #2 (Remarks to the Author):

Chen et al, present cryo-EM structures of the ABCD1 protein, which is the protein that is deficient in adrenoleukodystrophy (ALD).

Detailed information into the structure of the ABCD1 protein may provide insight in the molecular details underlying its function as a transporter from very long-chain fatty acids.

The authors report an impressive amount of data that provides interesting, new and important insight into the transport function of ABCD1. The function and structure of the ALD protein have been studied for over two decades with limited success. This manuscript adds valuable new insight into its structure and function. It is a very well written manuscript that reads very nicely. The figures are impressive. In my version a little fuzzy, but it is possible that these are not the final high-resolution versions.

The authors finish the introduction with: “these (cryo-EM) structures enabled us to precisely map the clinical mutations in ABCD1 gene, and interpret the molecular pathogenesis of X-ALD”. This is a claim that caught my interest instantly. However, I am neither an EM cryo, nor a modeling expert. Therefore, I cannot comment on the models themselves. I read many impressive observations like “Notably, in both substrate and ATP-bound forms, Arg280 from TM4 is stabilized by Asp194' and Glu199' from TM3' via salt bridges”. The manuscript has several of comparable sentences that are very interesting and informative for the readers and for the further interpretation of mutations. This is becoming more and more relevant now that countries are adding ALD to their newborn screening programs which requires interpretation of variants identified (in the absence of clinical clues). To me it is not really clear how such detailed info at the amino acid level can be obtained from the EM structures. Therefore, I hope the other reviewers can comment on this.

I have several comments, requests, suggestions

Introduction

Line 54-56: The authors state that ABCD1 also transports C18:1-CoA (refs 4 and 26). As far as I know C18:1 is a pure mitochondrially metabolized fatty acid and ABCD1 is a purely peroxisomal protein. As this is a very confusing and surprising claim I checked the refs. Ref 4 studies ABCD1, and Ref 26 both ABCD1 and ABCD2 (in a yeast model system). The authors in reference 26 write: “In the experiment of Fig. 5B we tested the oxidation of a number of unsaturated FA including C18:1, C18:2, C22:6 and C24:6. The results show no or minor restoration of oxidation capacity with C18:1 and C18:2 as substrate in pxa1/pxa2Δ cells expressing either HsABCD2 (ALDRP) alone or in combination with HsABCD1 (ALDP).” Therefore, this should be corrected.

Lines 56-58: The authors write that ABCD2 transports most of the substrates of ABCD1 and also poly-unsaturated fatty acids. While, there is a large body of studies and reports focused on ABCD1 function, there is a very limited number of studies on the function of ABCD2. The 2 papers the authors quote is a study in a yeast background and a study in the ABCD2 KO mouse. As of today, the exact substrate specificity of ABCD2 remains unclear. If the authors refer to over-expression studies than those results should be interpreted with some caution, because proteins behave and function different when over-expressed. At best there is some indication/evidence that ABCD2 transports C22:6-CoA and C24:6-CoA.

Results

At several occasions I miss a western blot and quantification of ABCD1 protein expressed. This is a crucial control for interpretation of the data.

For example:

Line 78: "As predicted, the expression level of this chimeric ABCD1 is ~15 folds to that of human ABCD1." What is missing here is a western blot showing this data. Especially as Fig 1b extended shows multiple lower weight bands for hABCD1 that may indicate break down products.

Lines 105-109: The C-terminal deletion mutant has an increased EC50 and sharply decreased Vmax. What is the effect of the deletion on residual protein expression? Less protein present affects the interpretation of the enzymatic activity readings.

Line 150, the authors mention a R401Q mutation. Fig 3G however shows a R401A and not the R401Q mutation. Which of the 2 is the correct annotation? The authors indeed indicate the R401Q is a mutation that has been identified in >40 patients and >20 publications. Therefore, it is important and relevant to know if this was a typo or they modeled R401A.

Overall: enzymatic activity data should be accompanied by protein expression data. The majority of missense mutations in ABCD1 affect protein stability (including missense mutations in R104 and R152). Therefore, differences in enzymatic activity can also be explained simply by differences in protein abundance due to the effect of the amino acid substitution on protein stability and level and not necessarily only due to altered amino acid residue. If substitution do affect protein stability and hence enzymatic activity the interpretation of the findings with respect to the role of a particular amino acid residue is very different.

Line 161-162: "To investigate what happens to ABCD1 succeeding substrate release, we introduced an E630Q mutation, which abolishes the ATP hydrolysis activity, but maintains the ATP-binding capacity." This observation misses either supporting data or a reference that supports this claim. Please provide a source. I don't see this mutation listed in the ABCD1 mutation database that the authors refer to.

Discussion

Extended Figure 8 in which the authors plotted 145 pathogenic missense mutations looks really nice. It would be very informative if they could, somehow, indicate the amino acid numbers in the figure. Now they write "mutation of Gly266, which is localized at the kink of TM4", but I have no idea which of the colored dots they are referring to. In light of the final sentence of the introduction "these (cryo-EM)

structures enabled us to precisely map the clinical mutations in ABCD1 gene, and interpret the molecular pathogenesis of X-ALD” a clearer and more informative figure would be very much appreciated.

Reviewer #3 (Remarks to the Author):

In this manuscript, Chen et al. describe molecular structures of a human peroxisomal ABC transporter ABCD1 in three conformational states (apo, substrate C22:0-CoA bound, and ATP bound). The study shows that this half-transporter of D family can form homodimer and bind to VLCFA-CoAs. The specific substrates stimulate the ATPase activity by >150 fold compared to the basal activity of ABCD1. The authors exhibit that the crossover helices formed by C-terminus from the two halves of the transporter are important for the ATPase activity of ABCD1 and may be involved in dimerization. The manuscript further establishes the importance of the amino acid residues involved in the substrate binding by comparing ATPase assays of various site-directed mutants. Although overall structure of chimeric human ABCD1 is very similar to previously published human ABCD1 (ATP bound and substrate bound structures) (Wang et.al, Cell Res. 2021) however, in the present study C22:0-CoA is used as substrate instead of oleoyl-CoA.

1. In the year 2021, there are 4 bioRxiv preprint entries on first human ABCD1 structures including one from the authors of this manuscript (<https://doi.org/10.1101/2021.09.24.461565>, <https://doi.org/10.1101/2021.09.04.458904>, <https://doi.org/10.1101/2021.10.14.464310>, <https://doi.org/10.1101/2021.09.24.461756>). One of these appeared in Cell Research early in November 2021 (<https://doi.org/10.1038/s41422-021-00585-8>) with two structures- human ABCD1EQ in oleoyl-CoA bound and ATP-bound conformations (PDB IDs: 7SHM and 7SHN). There is no mention of these publications in this manuscript. It would be great if authors can compare the results to the structures in the literature and add relevant references. Peroxisomal fatty acid metabolism is a physiologically important process. The ABCD1-3 transporters located on the peroxisomal membranes are the points of entry of various fatty acyl-CoAs. The readers would greatly benefit from the comparison and an elaborate discussion on the unique findings of this manuscript. The AlphaFold predicted structure of ABCD1 (AF-P33897-F1) overall has reasonable confidence. A brief comment on the validation of predicted structure through the experimental data in this manuscript would also be helpful.

2. CryoEM sample preparation section is missing in the methods section. Elaborate on the type of cryoEM grids used, concentration and volume of different samples applied, blotting time, humidity, instrument used to freeze grids. Elaborate on- How much substrate was added for the substrate-bound structure and how long the substrate was incubated? For ATP-bound structure how much ATP was added and incubated for how long and what temperature?

3. Extended Data Fig. 1 figure legend says that “The peak fractions of 10 mL for human ABCD1 and 10.2 mL for chimeric ABCD1 were pooled and concentrated for biochemical and structural studies. It is not clear in the manuscript though which experiments human ABCD1 and chimeric ABCD1 were used. It would be helpful to change the nomenclature to something like chABCD1 for the chimeric version throughout the manuscript and hABCD1 for full-length human ABCD1 protein. If human ABCD1 was used in any experiment, authors should comment on the purity of the protein. There are many contaminating bands of comparable strengths to the band marked as human ABCD1 in the gel (Extended Data Fig. 1b) while chimeric ABCD1 has higher purity level.

4. The purification profile/sizing peak of different mutants used in the study of ABCD1 should be included in the supplementary data.

5. Figure 1b- It is recommended to include the concentration dependent ATPase activity for Acetyl-CoA along with the specific substrates. This would further support the Acetyl-CoA is not a specific substrate at higher concentrate as well.

6. The authors mention that E630Q mutation abolishes the ATPase activity but retains the ATP-binding however no ATPase assay data was provided for this mutant in the manuscript. In the purification conditions used here, add ATPase activity comparison of WT vs E630Q protein in the manuscript.

7. The substrates were dissolved in 5% (w/v) methyl- β -cyclodextrin. Did the authors perform any control experiments on the effect of the solvent on the ATPase activity of ABCD1?

8. The structures are solved in the presence of digitonin detergent. Did authors perform any ATPase activity experiment in digitonin? It is suggested that a comparison between activity of WT ABCD1 in LMNG/CHS vs digitonin would be useful to know how the detergents impact the activity in case of ABCD1.

9. Comment on possible reason for the disappearance of C-terminus crossover in the substrate bound structure.

10. The authors interestingly show that the deletion of C-terminus adversely effects the ATPase activity (decrease in V_{max} by ~ 11 fold in the presence of the substrate) and they speculate that the C-terminus crossover may facilitate the dimerization of two NBDs. The authors also mention one of the clinically relevant mutations T693M in the C-terminus. The author should perform the ATPase activity with

T693M mutation. It will be significant for the community and will provide crucial insight on the specific role of T693M mutation in ABCD1 function and hence X-ALD disease.

11. Line 133- The adenine ring is stabilized by Lys213 and Ser213 from TM3. Update the correct amino acid number for these residues.

12. Line 150- the text says “portion (R104A, R152A, K217A, K336A, Y337F, and R401Q)” while in Figure 3g X-axis the mutation is R401A. Clarify.

13. Elaborate more on the rationale for mutant design for various residues used in the manuscript (e.g., A247W, G343V, P350W, and A395W).

14. Why do the authors think the PE density is present in the apo- ABCD1 structure and not the others?

15. Was the substrate density symmetric before imposing C2 symmetry in the cryo-EM data processing of substrate bound ABCD1 sample?

Reviewer #4 (Remarks to the Author):

In this work, Chen and coworkers describe 3 EM structures of “human” ABCD1. The structures reveal the transporter in an apo inward-facing conformation, substrate-bound inward-facing conformation, and an outward-facing ATP-bound conformation. Although the general details of the structures are very similar to those reported for homologous transporters, they reveal some novel features. Of special interest is the cross over of the 2 substrate molecules that essentially form an inter-domain bridge. While this work is of high quality, the authors should do a better job in placing their findings in the context of the available structural information of ABCD1 and similar ABC transporters

Major concerns

1. Recently, a paper describing the EM structure of ABCD1 was published (<https://www.nature.com/articles/s41422-021-00585-8>). The authors should explain what novel information their work contributes.

2. In both title and abstract the reader is led to believe that the structure was determined for human ABCD1, while in fact it was determined for a chimera. This should be made clear, by removing “human” from the title and mentioning the chimera in the abstract.

3. The apo-inward conformation and the ATP bound conformation must be aligned to structures of homologous ABC transporters. These alignments need to be shown in the main figures, with clear explanations of the similarities and differences.

4. The authors should generate mutant T693M and test their hypothesis regarding this mutant and ATPase activity of ABCD1.

5. Mutational analysis of C22:0-CoA residues: The authors propose that these results support the suggestion that these residues are indeed involved in ligand binding. However, unless they show that the K_d (or in their case apparent K_m) changes while the V_{max} did not, not such claim can be made. It is entirely possible that the mutation caused a structural perturbation that decreased the overall rate of ATP hydrolysis, irrespective of ligand binding.

6. For all bar figures the use of unpaired t test is wrong. The authors need to use ANOVA.

Additional comments

7. Figure 1a: was this assay performed in detergent, liposomes, Nanodiscs? This information must be given in the figure legend, along with concentrations of the protein and ligands.

8. The “unsharp map”? I am not familiar with this terminology.

9. Proposed model: is this model any different from those previously proposed for similar ABC transporters? To me, it seems almost identical. Please include a clear comparison.

10. “ Short-chain and medium-chain fatty acyl-CoAs can directly diffuse into mitochondria, whereas long-chain fatty acyl-CoAs (LCFA-CoAs) are transported to the mitochondria via the carnitine shuttle^{1,3}. In contrast, very long-chain fatty acyl-CoAs (VLCFA-CoAs) and branched-chain fatty acyl-CoA are respectively transported into the peroxisomes by three ATP-binding cassette (ABC)”

Please tell the reader what the lengths are of the short, medium, long (etc...) fatty acyl-CoAs

11. “and interpret the molecular pathogenesis of X-ALD”.

What exactly is “molecular pathogenesis” ? Perhaps the authors are referring to the molecular basis of pathogenesis?

12. “we introduced an E630Q mutation, which abolishes the ATP hydrolysis activity, but maintains the ATP-binding capacity”

Has this been verified by the authors? If so, these data need to be shown, or a reference needs to be provided.

Grammar:

“displayed more or less, but significant, decrease of ATPase activity in response to the addition of C22:0-CoA”

“As expected, despite 0.5 mM C22:0-CoA was added prior to the addition of 20 mM ATP/Mg²⁺”

Reviewer #1 (Remarks to the Author):

ABCD1 is a type IV ATP-binding cassette (ABC) transporter that catalyzes the translocation of long-chain and very-long-chain fatty acids (VLCFAs) into peroxisomes. Dysfunctional ABCD1 is the root cause of adrenoleukodystrophy (ALD), a rare disease affecting the nervous system and adrenal glands.

Chen et al. present three cryo-EM reconstructions of detergent-solubilized homodimeric ABCD1 (chimera of human ABCD1 containing N-terminal segment of *C. elegans* ABCD1) in the apo, substrate-bound, and ATP-bound states at overall resolutions of 3.5, 3.6, and 2.8 Å, respectively. First, the authors demonstrate that their recombinantly expressed chimera can catalyze ATP hydrolysis and that VLCFA-CoAs stimulate the ATPase activity of ABCD1. The apo structure exhibits the typical inward-facing conformation of type IV systems. An interesting feature of the apo state are dimerized C-terminal helices whose truncation leads to a significantly lower V_{max} , suggesting that these helices facilitate NBD dimerization and ATP hydrolysis, similar to the TAP-related heterodimeric transporter TmrAB (add ref. doi:10.1073/pnas.1620009114). This notion is corroborated by a disease-associated mutation in the helices. Interestingly, the substrate-bound structure shows that binding of C22:0-CoA induces a rigid-body movement of the TMDs, with the NBDs moving closer together and dissolving the interaction of the C-terminal helices. These conformational changes are accompanied by the formation of a short helix on the peroxisomal side of the transporter, similar to a helix in PglK (ref. 34). Chen et al. identify two molecules of C22:0-CoA in their map, and each molecule traverses across the two TMDs. The binding sites observed in the cryo-EM maps are confirmed by ATPase measurements with mutants harboring changes in the substrate-coordinating residues. Interestingly, two of the substrate-coordinating residues have also been described to be linked to ALD. Finally, the ATP-bound state adopts the typical outward-facing conformation and, similar to other type IV ABC transporters, most likely represents the stage after substrate release, as the substrate-binding sites are destroyed in this conformation.

The work by Chen et al. defines the structural basis of substrate binding and specificity of ABCD1 and outlines the conformational changes associated with different stages of the translocation cycle. Moreover, the authors map disease-causing mutations onto their ABCD1 structures and attempt to rationalize their pathogenic effects.

This work delivers significant new insights into the translocation mechanism of an important subfamily of ABC transporters, which warrant publication in Nature Communications. Although reasonably well written, it should be proofread again because it contains quite a few syntax/grammar/spelling mistakes throughout the text. The authors should also address the following points before publication:

Major points:

1) Fig. 3e: Some of the interaction distances (dashed lines) appear to be quite large. What cutoff did the authors choose to define interactions (hydrogen bonds and salt bridges)? Some dashed lines also do not seem to depict the interactions with the shortest distance (i.e., the strongest interactions), e.g.,

for R104.

A: Thank you for your suggestion. The cutoff distance we choose is ~3.5 Å for hydrogen bonds and ~4.0 Å for salt bridges. We have carefully verified the hydrogen bonds and salt bridges in Fig. 3e, and revised the figure accordingly after removing the mis-labeled hydrogen bond with Q332.

2) Some aspects of the apo and substrate-bound models should be improved: (i) apo: rotamers (and possibly Ramachandran outliers), (ii) substrate-bound: bond angles, clashscore, and rotamers.

A: Thank you for your comments. We have refined our models and updated the statistics as shown in the revised Supplementary Table 1.

3) The authors should mention the final size of the purified construct used for cryo-EM analysis. What will happen to the chimeric N-terminus? Is it processed further? The mature N-terminus of the construct should be determined by Edman sequencing.

A: The theoretical molecular weight of chABCD1 monomer is ~88 kDa, which is in agreement with the band in SDS-PAGE (Supplementary Fig. 1e). The recombinant protein was purified by anti-Flag gel, and the Flag-tag was fused to the most N-terminus, which was proved by Western blot (Supplementary Fig. 1b); thus, the sample applied to cryo-EM analysis is the full length chABCD1. However, the density of the N-terminus is missing probably due to its flexibility, similar to the recently reported human ABCD1 structures by other groups

(<https://doi.org/10.1101/2021.09.24.461565>, <https://doi.org/10.1101/2021.10.14.464310>, <https://doi.org/10.1101/2021.09.24.461756>, <https://doi.org/10.1038/s41422-021-00585-8>).

4) For the general readership, it would be helpful to mention in the introduction that ABCD1-4 are classified as type IV ABC transporters based on their structure and transmembrane fold (ref. doi:10.1146/annurev-biochem-011520-105201; doi:10.1002/1873-3468-13936).

A: We have added this information in the revised manuscript and related references.

5) Lines 207-210: The authors might consider removing the term “type II ABC exporter”, as they correctly point out that transporters with homologous TMDs include importers; thus, “type IV and type V ABC transporters” might indeed be the more appropriate term. => see also lines 327/328.

A: Corrected.

6) Line 160: The proof that substrates are translocated and released upon ATP binding have experimentally been demonstrated only recently for type IV transporter by single turnover experiments (consider ref. doi:10.7554/eLife.55943; doi:10.7554/eLife.67732 instead of citing a review).

A: We have added this information and related references.

7) Line 162: The authors might consider that an E630Q does not abolish the ATP hydrolysis. If the authors were to use a more sensitive assay, a slowdown of ATP turnover would be observed (again, see ref. doi:10.7554/eLife.55943).

A: Thank you for your suggestion. We have performed the ATPase activity assays of chABCD1-E630Q, in the presence or absence of 2 μM C22:0-CoA. In both cases, the activity of E630Q mutant is not detectable, which was shown in the Supplementary Fig. 7a, 7b. However, we have

revised the description according to your suggestion, and added the related reference.

8) In the abstract, the authors state that C22:0-CoA binds cooperatively to the transmembrane domains. However, no experimental evidence is provided for this statement. The Hill coefficient of 1.9 in substrate-stimulated ATPase activity likely reflects ATP binding and hydrolysis at the two nucleotide-binding sites.

A: Thank you for your comment. Indeed, the Hill coefficient of 1.9 in substrate-stimulated ATPase activity might also come from the two nucleotide-binding sites. However, as shown in Fig. 1a, the ATPase activity against ATP concentrations revealed a Michaelis-Menten curve. Thus there is no cooperativity between the two nucleotide-binding sites.

Minor points:

1) In the abstract: Please provide the full name of fatty acyl-coenzyme A (C22:0-CoA).

A: Provided.

2) Line 21: Consider removing “clearly” as illustrated snapshots should already be defined.

A: Removed.

3) Line 31: “Cytoplasm” is the wrong term as it includes the interior of the cell except the nucleus, so organelles such as peroxisomes are part of the cytoplasm. “Cytosol” would be the correct term. Please check carefully.

A: Corrected.

4) Line 47, 49, 50, etc.: “ABCD1-4 or ABCD1-3” instead of “ABCD1~4 or ABCD1~3”.

A: Corrected.

5) Line 15, 61: “cryogenic electron microscopy” instead of “cryo-electron microscopy”.

A: Corrected.

6) Line 146: “might provide” instead of “might provides”.

A: Corrected.

7) Line 224: “approach each other” instead of “approach towards each other”.

A: Corrected.

8) Lines 226/227: It is phosphate release after ATP hydrolysis rather than ATP hydrolysis itself that resets the transporter to the inward-facing state (ref. doi:10.1038/s41586-019-1391-0; doi:10.7554/eLife.55943).

A: Corrected.

9) Line 500: “maps” instead of “map”

A: Corrected.

Reviewer #2 (Remarks to the Author):

Chen et al, present cryo-EM structures of the ABCD1 protein, which is the protein that is deficient in adrenoleukodystrophy (ALD).

Detailed information into the structure of the ABCD1 protein may provide insight in the molecular details underlying its function as a transporter from very long-chain fatty acids.

The authors report an impressive amount of data that provides interesting, new and important insight into the transport function of ABCD1. The function and structure of the ALD protein have been studied for over two decades with limited success. This manuscript adds valuable new insight into its structure and function. It is a very well written manuscript that reads very nicely. The figures are impressive. In my version a little fuzzy, but it is possible that these are not the final high-resolution versions.

The authors finish the introduction with: “these (cryo-EM) structures enabled us to precisely map the clinical mutations in ABCD1 gene, and interpret the molecular pathogenesis of X-ALD”. This is a claim that caught my interest instantly. However, I am neither an EM cryo, nor a modeling expert. Therefore, I cannot comment on the models themselves. I read many impressive observations like “Notably, in both substrate and ATP-bound forms, Arg280 from TM4 is stabilized by Asp194' and Glu199' from TM3' via salt bridges”. The manuscript has several of comparable sentences that are very interesting and informative for the readers and for the further interpretation of mutations. This is becoming more and more relevant now that countries are adding ALD to their newborn screening programs which requires interpretation of variants identified (in the absence of clinical clues). To me it is not really clear how such detailed info at the amino acid level can be obtained from the EM structures. Therefore, I hope the other reviewers can comment on this.

I have several comments, requests, suggestions

Introduction

Line 54-56: The authors state that ABCD1 also transports C18:1-CoA (refs 4 and 26). As far as I know C18:1 is a pure mitochondrially metabolized fatty acid and ABCD1 is a purely peroxisomal protein. As this is a very confusing and surprising claim I checked the refs. Ref 4 studies ABCD1, and Ref 26 both ABCD1 and ABCD2 (in a yeast model system). The authors in reference 26 write: “In the experiment of Fig. 5B we tested the oxidation of a number of unsaturated FA including C18:1, C18:2, C22:6 and C24:6. The results show no or minor restoration of oxidation capacity with C18:1 and C18:2 as substrate in *pxa1/pxa2Δ* cells expressing either HsABCD2 (ALDRP) alone or in combination with HsABCD1 (ALDP).” Therefore, this should be corrected.
A: Corrected. We are sorry for the mistake.

Lines 56-58: The authors write that ABCD2 transports most of the substrates of ABCD1 and also poly-unsaturated fatty acids. While, there is a large body of studies and reports focused on ABCD1 function, there is a very limited number of studies on the function of ABCD2. The 2 papers the

authors quote is a study in a yeast background and a study in the ABCD2 KO mouse. As of today, the exact substrate specificity of ABCD2 remains unclear. If the authors refer to over-expression studies than those results should be interpreted with some caution, because proteins behave and function different when over-expressed. At best there is some indication/evidence that ABCD2 transports C22:6-CoA and C24:6-CoA.

A: Sorry for the inaccurate statements. We revised our statement as “Despite the previous reports indicated that ABCD2 might transport polyunsaturated fatty acyl-CoAs in yeast and mouse, the substrate specificity of human ABCD2, which shares a sequence identity of 62% with ABCD1, remains unclear”, and added related references accordingly.

Results

At several occasions I miss a western blot and quantification of ABCD1 protein expressed. This is a crucial control for interpretation of the data.

For example:

Line 78: “As predicted, the expression level of this chimeric ABCD1 is ~15 folds to that of human ABCD1.” What is missing here is a western blot showing this data. Especially as Fig 1b extended shows multiple lower weight bands for hABCD1 that may indicate break down products.

A: Thank you for your advice. We have performed the Western blot assays of the wild type and mutants, which are shown in the Supplementary Fig. 1b.

Lines 105-109: The C-terminal deletion mutant has an increased EC50 and sharply decreased Vmax. What is the effect of the deletion on residual protein expression? Less protein present affects the interpretation of the enzymatic activity readings.

A: Deletion of C-terminus slightly lowers the expression of residual protein of about 10%. However, all protein samples applied to the *in vitro* enzymatic activity assays were normalized. The expression levels of all versions were estimated by the Western blot, as shown in Fig. 1b.

Line 150, the authors mention a R401Q mutation. Fig 3G however shows a R401A and not the R401Q mutation. Which of the 2 is the correct annotation? The authors indeed indicate the R401Q is a mutation that has been identified in >40 patients and >20 publications. Therefore, it is important and relevant to know if this was a typo or they modeled R401A.

A: It should be R401Q in Fig 3G. Sorry for the typo mistake.

Overall: enzymatic activity data should be accompanied by protein expression data. The majority of missense mutations in ABCD1 affect protein stability (including missense mutations in R104 and R152). Therefore, differences in enzymatic activity can also be explained simply by differences in protein abundance due to the effect of the amino acid substitution on protein stability and level and not necessarily only due to altered amino acid residue. If substitution do affect protein stability and hence enzymatic activity the interpretation of the findings with respect to the role of a particular amino acid residue is very different.

A: According to your suggestion, we have performed the Western blot assays and the thermal stability assays of all ABCD1 variants shown in Supplementary Fig. 1b, 5e, 6c, respectively. Accordingly, all protein samples applied to the *in vitro* enzymatic activity assays were normalized;

and moreover, all mutants displayed a comparable thermal stability to the wild type.

Line 161-162: “To investigate what happens to ABCD1 succeeding substrate release, we introduced an E630Q mutation, which abolishes the ATP hydrolysis activity, but maintains the ATP-binding capacity.” This observation misses either supporting data or a reference that supports this claim. Please provide a source. I don’t see this mutation listed in the ABCD1 mutation database that the authors refer to.

A: Thank you for your suggestion. We have performed the ATPase activity assays of chABCD1-E630Q, in the presence or absence of 2 μ M C22:0-CoA. In both cases, the activity of E630Q mutant is not detectable, which was shown in the Supplementary Fig. 7b. We have also revised the description, and added the related reference.

Discussion

Extended Figure 8 in which the authors plotted 145 pathogenic missense mutations looks really nice. It would be very informative if they could, somehow, indicate the amino acid numbers in the figure. Now they write “mutation of Gly266, which is localized at the kink of TM4”, but I have no idea which of the colored dots they are referring to. In light of the final sentence of the introduction “these (cryo-EM) structures enabled us to precisely map the clinical mutations in ABCD1 gene, and interpret the molecular pathogenesis of X-ALD” a clearer and more informative figure would be very much appreciated.

A: Supplementary Fig. 8 was revised according to your suggestion.

Reviewer #3 (Remarks to the Author):

In this manuscript, Chen et al. describe molecular structures of a human peroxisomal ABC transporter ABCD1 in three conformational states (apo, substrate C22:0-CoA bound, and ATP bound). The study shows that this half-transporter of D family can form homodimer and bind to VLCFA-CoAs. The specific substrates stimulate the ATPase activity by >150 fold compared to the basal activity of ABCD1. The authors exhibit that the crossover helices formed by C-terminus from the two halves of the transporter are important for the ATPase activity of ABCD1 and may be involved in dimerization. The manuscript further establishes the importance of the amino acid residues involved in the substrate binding by comparing ATPase assays of various site-directed mutants. Although overall structure of chimeric human ABCD1 is very similar to previously published human ABCD1 (ATP bound and substrate bound structures) (Wang et.al, Cell Res. 2021) however, in the present study C22:0-CoA is used as substrate instead of oleoyl-CoA.

1. In the year 2021, there are 4 bioRxiv preprint entries on first human ABCD1 structures including one from the authors of this manuscript (<https://doi.org/10.1101/2021.09.24.461565>, <https://doi.org/10.1101/2021.09.04.458904>, <https://doi.org/10.1101/2021.10.14.464310>, <https://doi.org/10.1101/2021.09.24.461756>). One of these appeared in Cell Research early in November 2021 (<https://doi.org/10.1038/s41422-021-00585-8>) with two structures- human ABCD1EQ in oleoyl-CoA bound and ATP-bound conformations (PDB IDs: 7SHM and 7SHN). There is no mention of these publications in this manuscript. It would be great if authors can compare the results to the structures in the literature and add relevant

references. Peroxisomal fatty acid metabolism is a physiologically important process. The ABCD1-3 transporters located on the peroxisomal membranes are the points of entry of various fatty acyl-CoAs. The readers would greatly benefit from the comparison and an elaborate discussion on the unique findings of this manuscript. The AlphaFold predicted structure of ABCD1 (AF-P33897-F1) overall has reasonable confidence. A brief comment on the validation of predicted structure through the experimental data in this manuscript would also be helpful.

A: According to your suggestions, we performed the structural comparisons with other reported ABCD1 structures and the structures predicted by AlphaFold2 (Supplementary Fig. 9). A couple of sentences were added in the Discussion part.

2. CryoEM sample preparation section is missing in the methods section. Elaborate on the type of cryoEM grids used, concentration and volume of different samples applied, blotting time, humidity, instrument used to freeze grids. Elaborate on- How much substrate was added for the substrate-bound structure and how long the substrate was incubated? For ATP-bound structure how much ATP was added and incubated for how long and what temperature?

A: The information was added in the Method section.

3. Extended Data Fig. 1 figure legend says that “The peak fractions of 10 mL for human ABCD1 and 10.2 mL for chimeric ABCD1 were pooled and concentrated for biochemical and structural studies. It is not clear in the manuscript though which experiments human ABCD1 and chimeric ABCD1 were used. It would be helpful to change the nomenclature to something like chABCD1 for the chimeric version throughout the manuscript and hABCD1 for full-length human ABCD1 protein. If human ABCD1 was used in any experiment, authors should comment on the purity of the protein. There are many contaminating bands of comparable strengths to the band marked as human ABCD1 in the gel (Extended Data Fig. 1b) while chimeric ABCD1 has higher purity level.

A: According to your suggestion, human ABCD1 and the chimeric ABCD1 are termed hABCD1 and chABCD1, respectively. Notably, the N-terminal segment from *C. elegans*, which was proposed to be involved in their subcellular location, is not traceable in our present structure. In fact, the counterpart N-terminal region is not structured in other reported human ABCD1 structures (<https://doi.org/10.1101/2021.09.24.461565>, <https://doi.org/10.1101/2021.10.14.464310>, <https://doi.org/10.1101/2021.09.24.461756>, <https://doi.org/10.1038/s41422-021-00585-8>). And only chABCD1 was applied to further activity assays and structure determination.

4. The purification profile/sizing peak of different mutants used in the study of ABCD1 should be included in the supplementary data.

A: Thank you for the suggestion. We added the purification profile/sizing peak in Supplementary Fig. 5d, 6b, 7a.

5. Figure 1b- It is recommended to include the concentration dependent ATPase activity for Acetyl-CoA along with the specific substrates. This would further support the Acetyl-CoA is not a specific substrate at higher concentration as well.

A: According to your recommendation, we have performed the concentration dependent ATPase activity for acetyl-CoA along with the specific substrates, which further supported that acetyl-CoA is not a specific substrate, even at the higher concentration (Fig. 1b).

6. The authors mention that E630Q mutation abolishes the ATPase activity but retains the ATP-binding however no ATPase assay data was provided for this mutant in the manuscript. In the purification conditions used here, add ATPase activity comparison of WT vs E630Q protein in the manuscript.

A: Thank you for your suggestion. We have performed the ATPase activity assays of chABCD1-E630Q, in the presence or absence of 2 μ M C22:0-CoA. In both cases, the activity of E630Q mutant is not detectable, which was shown in Supplementary Fig. 7b. Accordingly, we have revised the description according to your suggestion, and added the related reference.

7. The substrates were dissolved in 5% (w/v) methyl- β -cyclodextrin. Did the authors perform any control experiments on the effect of the solvent on the ATPase activity of ABCD1?

A: We performed the ATPase activity assays in the presence of 0.05% (w/v) methyl- β -cyclodextrin (Fig. 1a), which revealed no effect on the ATPase activity of ABCD1.

8. The structures are solved in the presence of digitonin detergent. Did authors perform any ATPase activity experiment in digitonin? It is suggested that a comparison between activity of WT ABCD1 in LMNG/CHS vs digitonin would be useful to know how the detergents impact the activity in case of ABCD1.

A: According to your suggestion, we performed the ATPase activity assays in digitonin, and compared with that in LMNG/CHS (Supplementary Fig. 1e). It indicated that the activity of ABCD1 in digitonin is much lower than that in LMNG/CHS.

9. Comment on possible reason for the disappearance of C-terminus crossover in the substrate bound structure.

A: We suppose that the conformational changes of NBDs upon substrate binding might break the interactions between the two C-terminal helices, which become highly flexible.

10. The authors interestingly show that the deletion of C-terminus adversely effects the ATPase activity (decrease in V_{max} by \sim 11 fold in the presence of the substrate) and they speculate that the C-terminus crossover may facilitate the dimerization of two NBDs. The authors also mention one of the clinically relevant mutations T693M in the C-terminus. The author should perform the ATPase activity with T693M mutation. It will be significant for the community and will provide crucial insight on the specific role of T693M mutation in ABCD1 function and hence X-ALD disease.

A: We performed the ATPase activity assays of T693M mutation. The gel filtration and SDS-PAGE profile, as well as the thermal stability assays indicated that the T693M mutant is well-folded, as shown in Supplementary, Fig. 5c, 5d. However, the ATPase activity assays revealed it possesses an almost non-detectable activity, as shown in Fig. 2c.

11. Line 133- The adenine ring is stabilized by Lys213 and Ser213 from TM3. Update the correct amino acid number for these residues.

A: Updated.

12. Line 150- the text says “portion (R104A, R152A, K217A, K336A, Y337F, and R401Q)” while in Figure 3g X-axis the mutation is R401A. Clarify.

A: Corrected. Sorry for the typo mistake.

13. Elaborate more on the rationale for mutant design for various residues used in the manuscript (e.g., A247W, G343V, P350W, and A395W).

A: Thank you for your comments. The putative substrate-binding residues A247, P350 and A395 were mutated to Trp, which might hinder the substrate binding. Indeed, these mutants possess an almost non-detectable activity (Fig. 1g). In addition, G343V is a disease related mutant.

14. Why do the authors think the PE density is present in the apo- ABCD1 structure and not the others?

A: PE was fitted based on the shape of density in the apo-ABCD1 structure. PE was most likely repelled in the substrate-bound and ATP-bound structures, due to steric hindrance with the substrate, or drastic conformational changes upon ATP binding, respectively.

15. Was the substrate density symmetric before imposing C2 symmetry in the cryo-EM data processing of substrate bound ABCD1 sample?

A: Yes, we observed two symmetric substrates without imposing symmetry.

Reviewer #4 (Remarks to the Author):

In this work, Chen and coworkers describe 3 EM structures of “human” ABCD1. The structures reveal the transporter in an apo inward-facing conformation, substrate-bound inward-facing conformation, and an outward-facing ATP-bound conformation. Although the general details of the structures are very similar to those reported for homologous transporters, they reveal some novel features. Of special interest is the cross over of the 2 substrate molecules that essentially form an inter-domain bridge. While this work is of high quality, the authors should do a better job in placing their findings in the context of the available structural information of ABCD1 and similar ABC transporters

Major concerns

1. Recently, a paper describing the EM structure of ABCD1 was published (<https://www.nature.com/articles/s41422-021-00585-8>). The authors should explain what novel information their work contributes.

A: According to your suggestions, we performed structural comparisons with the reported ABCD1 structures (Supplementary Fig. 9). Accordingly, a couple of sentences were added in the Discussion part.

2. In both title and abstract the reader is led to believe that the structure was determined for human ABCD1, while in fact it was determined for a chimera. This should be made clear, by removing “human” from the title and mentioning the chimera in the abstract.

A: According to your suggestion, human ABCD1 and the chimeric ABCD1 are termed hABCD1 and chABCD1, respectively. Notably, the N-terminal segment from *C. elegans*, which was proposed to be involved in their subcellular location, is not traceable in our present structure. In fact, the counterpart N-terminal region is not structured in other reported human ABCD1 structures (<https://doi.org/10.1101/2021.09.24.461565>, <https://doi.org/10.1101/2021.10.14.464310>, <https://doi.org/10.1101/2021.09.24.461756>, <https://doi.org/10.1038/s41422-021-00585-8>).

3. The apo-inward conformation and the ATP bound conformation must be aligned to structures of homologous ABC transporters. These alignments need to be shown in the main figures, with clear explanations of the similarities and differences.

A: According to your suggestion, we compared with the structures of some ABC homologues shown in Supplementary Fig. 5a, 7c. A couple of sentences were added in the revised manuscript

4. The authors should generate mutant T693M and test their hypothesis regarding this mutant and ATPase activity of ABCD1.

A: We performed the ATPase activity assays of T693M mutation. The gel filtration and SDS-PAGE profile, as well as the thermal stability assays indicated that the T693M mutant is well-folded. However, the ATPase activity assays revealed it possesses an almost non-detectable activity, as shown in Fig. 2c.

5. Mutational analysis of C22:0-CoA residues: The authors propose that these results support the suggestion that these residues are indeed involved in ligand binding. However, unless they show that the K_d (or in their case apparent K_m) changes while the V_{max} did not, not such claim can be made. It is entirely possible that the mutation caused a structural perturbation that decreased the overall rate of ATP hydrolysis, irrespective of ligand binding.

A: According to your suggestion, we have measured V_{max} and EC₅₀ of the three mutants of higher activity (Supplementary Fig. 6d, 6e). In fact, the other 7 mutants displayed too low activity to detect the catalytic parameters. The P350W mutant displayed a reduced V_{max}, whereas two mutants R152A and Y337F displayed a reduced V_{max} and an increased EC₅₀. However, the gel filtration and SDS-PAGE profiles, as well as the thermal stability assays indicated that the all these mutants are properly folded (Supplementary Fig. 6c). Therefore, combined with the structure data, we made the conclusion that these results support the suggestion that these residues are indeed involved in ligand binding.

6. For all bar figures the use of unpaired t test is wrong. The authors need to use ANOVA.

A: Corrected.

Additional comments

7. Figure 1a: was this assay performed in detergent, liposomes, Nanodiscs? This information must be given in the figure legend, along with concentrations of the protein and ligands.

A: The assay was performed in detergent, and we have added this information, along with concentrations of the protein and ligands in the figure legend.

8. The “unsharp map”? I am not familiar with this terminology.

A: Sorry for the typo mistake. It should be “unsharpened map”, which is the map generated by 3D-auto refine before being sharpened according to the B-factor.

9. Proposed model: is this model any different from those previously proposed for similar ABC transporters? To me, it seems almost identical. Please include a clear comparison.

A: Following your suggestions, the proposed model was slightly revised. Our model differs from the previous ones in two points: 1) cooperative binding of the substrates, and 2) a unique C-terminal crossover.

10. “Short-chain and medium-chain fatty acyl-CoAs can directly diffuse into mitochondria, whereas long-chain fatty acyl-CoAs (LCFA-CoAs) are transported to the mitochondria via the carnitine shuttle. In contrast, very long-chain fatty acyl-CoAs (VLCFA-CoAs) and branched-chain fatty acyl-CoA are respectively transported into the peroxisomes by three ATP-binding cassette (ABC)” Please tell the reader what the lengths are of the short, medium, long (etc...) fatty acyl-CoAs

A: We added this information the revised manuscript.

11. “and interpret the molecular pathogenesis of X-ALD”. What exactly is “molecular pathogenesis”? Perhaps the authors are referring to the molecular basis of pathogenesis?

A: Yes, we revised it to “the molecular basis of pathogenesis”.

12. “we introduced an E630Q mutation, which abolishes the ATP hydrolysis activity, but maintains the ATP-binding capacity” Has this been verified by the authors? If so, these data need to be shown, or a reference needs to be provided.

A: Thank you for your suggestion. We have performed the ATPase activity assays of chABCD1-E630Q, in the presence or absence of 2 μ M C22:0-CoA. In both cases, the activity of E630Q mutant is not detectable, which was shown in Supplementary Fig. 7b Accordingly, we have revised the description according to your suggestion, and added the related reference.

Grammar:

“displayed more or less, but significant, decrease of ATPase activity in response to the addition of C22:0-CoA”

A: Corrected.

“As expected, despite 0.5 mM C22:0-CoA was added prior to the addition of 20 mM ATP/Mg²⁺”

A: Corrected.

REVIEWERS' COMMENTS

Reviewer #1 (Remarks to the Author):

The comments and major concerns raised by the reviewers have been addressed accordingly. The overall conclusion have been strengthened in the revised manuscript.

Reviewer #2 (Remarks to the Author):

The authors have addressed all the questions and suggestions raised by the reviewers and have performed additional experiments to strengthen the work and improve the manuscript.

I have a few minor points remaining and one request/suggestion regarding the improved supplemental figure 8.

Lines 41-42: The sentence "A series of reports indicated that X-ALD is caused by mutations in the ABCD1 gene (ALDP) 5,13,14" needs a few subtle changes.

- In recent years, the term "mutation" is replaced by "pathogenic variant". A mutation indicates a change from the reference DNA sequence and this doesn't have to be disease-causing. In addition, the word "mutation" has a negative connotation. Please change "mutation" to "pathogenic variant" throughout the manuscript.

- Reference 14 describes the generation of the ALD knockout mouse and not the genetic cause for ALD. Please remove and replace by the following suggestion

- "A series of reports" The ABCD1 variant database (<https://adrenoleukodystrophy.info> PMID: 35053399) reports >940 distinct pathogenic variants in the ABCD1 gene and >200 publications of pathogenic variants. Please add this recent (2022) reference (Mallack et al 2022 PMID: 35053399) to the database instead of the wrong reference to the ALD knockout mouse publication.

- Please update the sentence to: "X-ALD is caused by pathogenic variants in the ABCD1 gene (ALDP)(Refs Wiesinger et al 2013; Mosser et al 1993; Mallack et al 2022)

Line 55: C26:1, please change to C26:1-CoA as all fatty acids are transported as CoA-esters.

Lines 212-215: Please change “mutations” to “pathogenic variants” and add reference Mallack et al 2022 PMID: 35053399.

Line 215: The authors state that they mapped 145 missense variants on the C22:0-CoA-bound ABCD1 structure (Supplementary Fig. 8). These are shown in the coloured dots in the figure. Upon request by the reviewer they indicated some (11/145) of the variants by adding the amino acid residue. From a clinical perspective, it would be very valuable to see all 145 missense variants being mapped. I realise the figure will be crowded. However, in accordance with the final sentence of the introduction which states “Moreover, these structures enabled us to precisely map the clinical mutations in ABCD1 gene, and interpret the molecular basis of pathogenesis of X-ALD” would it be possible to provide a little more information?

Maybe indicate in the legend which amino acid residues belong to the NBD, substrate-binding pocket and conformation coupling? Like, PEX19 binding amino acid residues (R74 – S98) I extracted this info from the figure.

In addition and in line with the previous remark.

In line 217 the authors write “... we found that up to 59 sites are located on the NBDs, including 15 residues directly participating in ATP-Mg²⁺ binding”, but it is not clear from the figure and not indicated in the text which residues these 15 are. Would it be possible to somehow provide this information? Maybe in a Table or the legend? Mg²⁺ binding: amino acid residues A, X, Y, and Z. This additional information that the authors have - but is not clearly shared in the manuscript - would be very much appreciated and it really would broaden the impact of this important work. Especially now that ALD is being added to newborn screening programs which results in more need regarding the pathogenicity of a variant identified.

Reviewer #3 (Remarks to the Author):

The authors have responded well to the reviewer comments and have made necessary changes in the manuscript. I have a few minor comments-

1. The sample preparation for C22:0-CoA-bound ABCD1 complex and ATP-bound ABCD1 complex is identical (page 16). Authors should confirm if this was intentional or correct it.
2. Figure 1a shows ATPase activity of WT ABCD1 protein in the presence of water in blue, 2 μ M C22:0-CoA in red, and, 0.05% methyl- β -cyclodextrin (M- β -CD) in black. I am assuming it is a typographical error. According to the current representation it looks like protein is most active in the presence of 0.05% M- β -CD (plot in black). Please fix this or respond if I am not understanding it correctly.

3. In general, many data points in the graphs throughout the manuscript do not have error bars (example- Figure 1a, 1b). Authors should confirm that all the assays were performed and reported in replicates (mention n=?).

4. It seems perplexing to me that when you delete the region Glu694-Thr745 the protein retains some ATPase activity as compared to the WT but when you mutate a single residue in that region (T693M) there is no apparent ATPase activity. A comment on that in the results section can help the audience.

Reviewer #4 (Remarks to the Author):

The authors have addressed my concerns

Reviewer #2 (Remarks to the Author):

The authors have addressed all the questions and suggestions raised by the reviewers and have performed additional experiments to strengthen the work and improve the manuscript.

I have a few minor points remaining and one request/suggestion regarding the improved supplemental figure 8.

1. Lines 41-42: The sentence “A series of reports indicated that X-ALD is caused by mutations in the ABCD1 gene (ALDP) 5,13,14” needs a few subtle changes.
- In recent years, the term “mutation” is replaced by “pathogenic variant”. A mutation indicates a change from the reference DNA sequence and this doesn’t have to be disease-causing. In addition, the word “mutation” has a negative connotation. Please change "mutation" to "pathogenic variant" throughout the manuscript.

A: Revised.

2. - Reference 14 describes the generation of the ALD knockout mouse and not the genetic cause for ALD. Please remove and replace by the following suggestion
- “A series of reports” The ABCD1 variant database (<https://adrenoleukodystrophy.info> PMID: 35053399) reports >940 distinct pathogenic variants in the ABCD1 gene and >200 publications of pathogenic variants. Please add this recent (2022) reference (Mallack et al 2022 PMID: 35053399) to the database instead of the wrong reference to the ALD knockout mouse publication.
- Please update the sentence to: “X-ALD is caused by pathogenic variants in the ABCD1 gene (ALDP)(Refs Wiesinger et al 2013; Mosser et al 1993; Mallack et al 2022)

A: Thank you for your suggestion. We have updated the statement and related references.

3. Line 55: C26:1, please change to C26:1-CoA as all fatty acids are transported as CoA-esters.

A: Revised.

4. Lines 212-215: Please change “mutations” to “pathogenic variants” and add reference Mallack et al 2022 PMID: 35053399.

A: We have revised “mutations” to “pathogenic variants”, and also updated the statement and related reference.

5. Line 215: The authors state that they mapped 145 missense variants on the C22:0-CoA-bound ABCD1 structure (Supplementary Fig. 8). These are shown in the coloured dots in the figure. Upon request by the reviewer they indicated some (11/145) of the variants by adding the amino acid residue. From a clinical perspective, it would be very valuable to see all 145 missense variants being

mapped. I realise the figure will be crowded. However, in accordance with the final sentence of the introduction which states “Moreover, these structures enabled us to precisely map the clinical mutations in ABCD1 gene, and interpret the molecular basis of pathogenesis of X-ALD” would it be possible to provide a little more information? Maybe indicate in the legend which amino acid residues belong to the NBD, substrate-binding pocket and conformation coupling? Like, PEX19 binding amino acid residues (R74 – S98) I extracted this info from the figure.

In addition and in line with the previous remark.

In line 217 the authors write “... we found that up to 59 sites are located on the NBDs, including 15 residues directly participating in ATP-Mg²⁺ binding”, but it is not clear from the figure and not indicated in the text which residues these 15 are. Would it be possible to somehow provide this information? Maybe in a Table or the legend? Mg²⁺ binding: amino acid residues A, X, Y, and Z. This additional information that the authors have - but is not clearly shared in the manuscript - would be very much appreciated and it really would broaden the impact of this important work. Especially now that ALD is being added to newborn screening programs which results in more need regarding the pathogenicity of a variant identified.

A: Thank you for your suggestion. As you have mentioned, it would be crowded to add all the information in the figure. Thus, we now label all the pathogenic variants in ABCD1 structure via dots of various colors (Supplementary Fig. 8), and list the corresponding information of these pathogenic variants in the Supplementary Table 2.

Reviewer #3 (Remarks to the Author):

The authors have responded well to the reviewer comments and have made necessary changes in the manuscript. I have a few minor comments-

1. The sample preparation for C22:0-CoA-bound ABCD1 complex and ATP-bound ABCD1 complex is identical (page 16). Authors should confirm if this was intentional or correct it.

A: Sorry for the mistake, which has been corrected.

2. Figure 1a shows ATPase activity of WT ABCD1 protein in the presence of water in blue, 2 μ M C22:0-CoA in red, and, 0.05% methyl- β -cyclodextrin (M- β -CD) in black. I am assuming it is a typographical error. According to the current representation it looks like protein is most active in the presence of 0.05% M- β -CD (plot in black). Please fix this or respond if I am not understanding it correctly.

A: Corrected. We are sorry for the typo.

3. In general, many data points in the graphs throughout the manuscript do not have error bars (example- Figure 1a, 1b). Authors should confirm that all the assays were performed and reported in replicates (mention n=?).

A: We have added all the necessary information in this revision

4. It seems perplexing to me that when you delete the region Glu694-Thr745 the protein retains some ATPase activity as compared to the WT but when you mutate a single residue in that region (T693M) there is no apparent ATPase activity. A comment on that in the results section can help the audience.

A: We suppose that the T693M mutation impairs the interactions between the two C-terminal helices, leading to the highly flexibility of the C-terminal helices, which somewhat interfere the dimerization of NBDs. However, ABCD1 with the truncated C-terminal helix still remain the capability of dimerization, thus retaining ~10% ATPase activity.